# DiffTraj: Generating GPS Trajectory with Diffusion Probabilistic Model

**Yuanshao Zhu**[1,2], **Yongchao Ye**[1], **Shiyao Zhang**[1], **Xiangyu Zhao**[2,*], **James J.Q. Yu**[3,*]

[1] Southern University of Science and Technology
[2] City University of Hong Kong
[3]University of York
{zhuys2019, 12032868}@mail.sustech.edu.cn
zhangsy@sustech.edu.cn
xianzhao@cityu.edu.hk
james.yu@york.ac.uk

## Abstract

Pervasive integration of GPS-enabled devices and data acquisition technologies has led to an exponential increase in GPS trajectory data, fostering advancements in spatial-temporal data mining research. Nonetheless, GPS trajectories contain personal geolocation information, rendering serious privacy concerns when working with raw data. A promising approach to address this issue is trajectory generation, which involves replacing original data with generated, privacy-free alternatives. Despite the potential of trajectory generation, the complex nature of human behavior and its inherent stochastic characteristics pose challenges in generating high-quality trajectories. In this work, we propose a spatial-temporal diffusion probabilistic model for trajectory generation (DiffTraj). This model effectively combines the generative abilities of diffusion models with the spatial-temporal features derived from real trajectories. The core idea is to reconstruct and synthesize geographic trajectories from white noise through a reverse trajectory denoising process. Furthermore, we propose a Trajectory UNet (Traj-UNet) deep neural network to embed conditional information and accurately estimate noise levels during the reverse process. Experiments on two real-world datasets show that DiffTraj can be intuitively applied to generate high-fidelity trajectories while retaining the original distributions. Moreover, the generated results can support downstream trajectory analysis tasks and significantly outperform other methods in terms of geo-distribution evaluations.

## 1 Introduction

GPS trajectory data play a crucial role in numerous spatial-temporal data mining applications, including urban traffic planning, business location selection, and travel time estimation [43, 42]. Despite the substantial progress in urban applications through GPS trajectory analysis, few attention was paid to data accessibility and privacy concerns [41]. For example, real-world trajectory data, which contain sensitive personal geolocation information, raise significant privacy issues when using directly [38, 44]. Additionally, the time-consuming and labor-intensive nature of trajectory data collection leads to challenges in obtaining accessible, publicly available datasets that preserve privacy. Therefore, it is essential to develop methods that facilitate the efficient use of trajectory data in urban applications while safeguarding personal privacy. One promising approach involves generating synthetic trajectories by learning from real trajectory distributions and replacing privacy-sensitive

---

*Corresponding author

37th Conference on Neural Information Processing Systems (NeurIPS 2023).

real trajectories with these synthetic counterparts. The generated trajectories allow for equivalent data analysis outcomes and support high-level decision-making without compromising privacy.

However, generating GPS trajectory that accurately reflect real-world distributions will encounter the following challenges in practice. First, most cities contain various regions with diverse functions and population densities, resulting in non-independent and identically distributed trajectories among these regions [44, 40]. This complex distribution makes it difficult to learn a global trajectory model that captures regional nuances. Second, the inherent stochastic nature of human activities makes each trajectory unique, posing challenges in modeling and predicting individual movement patterns [41, 16]. Third, external factors, such as traffic conditions, departure time, and local events, significantly influence personal travel schedules [36]. Accounting for these factors and understanding their impact on the correlations between individual GPS locations within a trajectory adds to the complexity of modeling trajectory generation accurately.

To address the above challenges, we propose a trajectory generation method based on the spatial-temporal diffusion probabilistic model, which can effectively capture the complex behaviors of real-world activities and generates high-quality trajectories. The core idea behind this method is to perturb the trajectory distribution with noise through a forward trajectory diffusion process and then recover the original distribution by learning the backward diffusion process (denoising), resulting in a highly flexible trajectory generation model [11, 28]. We propose this framework based on three primary motivations: (i) The diffusion model is a more reliable and robust method of generation than canonical methods [35]. (ii) Human activities in the real world exhibit stochastic and uncertain characteristics [18, 10], and the diffusion model reconstructs data step-by-step from random noise, making it suitable for generating more realistic GPS trajectory. (iii) Since the diffusion model generates trajectories from random noise, it eliminates the risk of privacy leakage.

While the diffusion model shows promise for generating high-quality trajectories, it is a non-trivial task when applied to GPS trajectory generation. Firstly, the spatial-temporal nature of trajectory data presents complexity to the diffusion process, requiring the model to consider the spatial distribution of GPS points and the temporal dependencies among them. Secondly, external factors such as traffic conditions, departure time, and regions can significantly impact human mobility patterns [44], posing challenges for modeling these factors within the diffusion framework. Finally, the inherently stochastic nature of human activities demands that the diffusion model be capable of capturing a wide range of plausible trajectory patterns, which can be difficult due to the need to balance diversity and realism in the generated trajectories. Addressing these challenges is crucial for successfully applying the diffusion model to GPS trajectory generation and ensuring privacy-preserving, accurate, and computationally efficient outcomes.

Building on the identified motivations and addressing the limitations of applying the diffusion model to trajectory generation, we propose a **Diff**usion probabilistic model **Traj**ectory generation (DiffTraj) framework. This framework can generate infinite trajectories while preserving the real-world trajectory distribution. DiffTraj incorporates spatial-temporal modeling of raw trajectories without the need for additional operations, allowing for direct application to trajectory generation tasks. Crucially, the synthetic trajectories generated by DiffTraj ensure high generation accuracy while maintaining their utility, offering a promising solution to privacy-preserving GPS trajectory generation. To summarize, the contributions of this work are concluded as follows:

- We propose a DiffTraj framework for trajectory generation, leveraging the diffusion model to simulate real-world trajectories while preserving privacy. In addition, this framework can be applied to generate high-fidelity trajectory data with high efficiency and transferability. To the best of our knowledge, our work is the first exploration of trajectory generation by the diffusion model.

- We design a novel denoising network structure called Traj-UNet, which integrates residual blocks for spatial-temporal modeling and multi-scale feature fusion to accurately predict the noise level during denoising. Meanwhile, Traj-UNet integrates Wide and Deep networks [6] to construct conditional information, enabling controlled trajectory generation.

- We validate the effectiveness of DiffTraj on two real-world trajectory datasets, showing that the proposed methods can generate high-fidelity trajectories and preserve statistical properties. At the same time, the generated trajectories can support downstream trajectory analysis tasks with replaceable utility and privacy-preserving.

## 2 Related Work

**Trajectory Data Synthesizing**. Existing methods for trajectory data can be generally divided into two main categories: non-generative and generative [4]. Non-generative methods include perturbing real trajectories [1, 37] or combining different real trajectories [21]. Adding random or Gaussian perturbations protects privacy but compromises data utility by altering the spatial-temporal characteristics and data distribution. Striking a balance between trajectory utility and privacy protection is challenging [4]. Although Mehmet et al. generated forged trajectories by mixing different trajectories, this method relies on massive data and sophisticated mixing processes [21].

Besides, the principle of the generative method is to leverage deep neural networks that learn the spatial-temporal distribution underlying the real data. New trajectory data are therefore generated by sampling from the learned distribution. Liu et al. proposed a preliminary solution that employs generative adversarial networks (GAN) for trajectory generation, yet it failed to go further towards a detailed design[33]. Subsequently, some works divided the city map into grids and performed trajectory generation by learning the distribution of trajectories among the grids [38, 24]. However, there is a trade-off between generation accuracy and grid size. Meanwhile, researchers used the image generation capability of GAN to convert the trajectory data into images for time-series generation [4, 31], while the transformation between images and trajectories imposed an additional computational burden. Compared with previous methods, DiffTraj uses the diffusion model for trajectory generation, which can better explore the spatial and temporal distributions without additional data manipulation.

**Diffusion Probabilistic Model**. The diffusion model is a probabilistic generative model, which was first proposed by Sohl-Dickstein et al. [27] and then further improved by Ho et al. [11] and Song et al. [29]. A typical diffusion model generates synthetic data via two sequential processes, i.e., a forward process that gradually perturbs the data distribution by adding noise on multiple scales and a reverse process that learns how to recover the data distribution[35]. In addition, researchers have made extensive attempts to improve generative sample quality and sampling speed. For example, Song et al. proposed a non-Markovian diffusion process to reduce the sampling steps [28], Nichol et al. proposed to learn the variance of reverse processes allowing fewer sampling steps [22], and Dhariwal et al. searched for the optimal structure of the reverse denoising neural network to obtain better sampling quality. As a new type of advanced generative model, diffusion models have achieved superior performance over alternative generative models in various generative tasks, such as computer vision [25], natural language processing [15], multi-modal learning [2, 19], and traffic forecasting [32]. Nevertheless, the diffusion probabilistic model calls for efforts in spatial-temporal trajectory data generation. To the best of our knowledge, this work is a pioneering attempt that uses diffusion probabilistic model to generate GPS trajectory.

## 3 Preliminary

In this section, we first introduce the definitions and notations we use in this paper and then briefly present the fundamentals of the diffusion probabilistic model.

### 3.1 Problem Definition

**Definition 1: (GPS Trajectory)**. A GPS trajectory is defined as a sequence of continuously sampled private GPS location points, denoted by $x = \{p_1, p_2, \ldots, p_n\}$. The $i$-th GPS point is represented as a tuple $p_i = [\text{lat}_i, \text{lng}_i]$, where $\text{lng}_i$ and $\text{lat}_i$ denote longitude and latitude, respectively.

**Problem Statement: (Trajectory Generation)**. Given a set of real-world GPS trajectories, $\mathcal{X} = \{x^1, x^2, \ldots, x^m\}$, where each $x^i = \{p_1^i, p_2^i, \ldots, p_n^i\}$ is a sequence of private GPS location points. The objective of the trajectory generation task is to learn a generative model, $G$, that can generate synthetic GPS trajectories, $\mathcal{Y} = \{y^1, y^2, \ldots, y^k\}$, where each $y^i = \{q_1^i, q_2^i, \ldots, q_n^i\}$. such that:

- **Similarity**: The generated trajectories $y^i$ preserve the spatial-temporal characteristics and distribution of the real trajectories $x^i$. In practice, there are various physical meanings, such as the regional distribution of trajectory points, the distance between successive points, etc.

- **Utility**: The generated trajectories $y^i$ maintain utility for downstream applications and analyses.

- **Privacy**: Privacy is preserved, meaning that the generated trajectories $y^i$ do not reveal sensitive information about the individuals associated with the real trajectories $x^i$.

## 3.2 Diffusion Probabilistic Model

The diffusion probabilistic model is a powerful and flexible generative model, which has gained increasing attention in recent years for its success in various data generation tasks [11, 28, 29]. In general, the diffusion probabilistic model consists of two main processes: a forward process that gradually perturbs the data distribution with noise, and a reverse (denoising) process that learns to recover the original data distribution.

**Forward process**. Given a set of real data samples $x_0 \sim q(x_0)$, the forward process adds $T$ time-steps of Gaussian noise $\mathcal{N}(\cdot)$ to it, where $T$ is an adjustable parameter. Formally, the forward process can be defined as a Markov chain from data $x_0$ to the latent variable $x_T$:

$$q(x_{1:T} \mid x_0) = \prod_{t=1}^{T} q(x_t \mid x_{t-1}); \quad q(x_t \mid x_{t-1}) = \mathcal{N}\left(x_t; \sqrt{1-\beta_t}x_{t-1}, \beta_t\mathbf{I}\right). \tag{1}$$

Here, $\{\beta_t \in (0,1)\}_{t=1}^{T}$ $(\beta_1 < \beta_2 < \ldots < \beta_T)$ is the corresponding variance schedule. Since it is impractical to back-propagate the gradient by sampling from a Gaussian distribution, we adopt a reparameterization trick to keep the gradient derivable [11] and the $x_t$ can be expressed as $x_t = \sqrt{\bar{\alpha}_t}x_0 + \sqrt{1-\bar{\alpha}_t}\epsilon$, where $\epsilon \sim \mathcal{N}(0, \mathbf{I})$ and $\bar{\alpha}_t = \prod_{i=1}^{t}(1-\beta_i)$.

**Reverse process**. The reverse diffusion process, also known as the denoising process, aims to recover the original data distribution from the noisy data $x_T \sim \mathcal{N}(0, \mathbf{I})$. Accordingly, this process can be formulated by the following Markov chain:

$$p_\theta(x_{0:T}) = p(x_T)\prod_{t=1}^{T} p_\theta(x_{t-1} \mid x_t); \quad p_\theta(x_{t-1} \mid x_t) = \mathcal{N}\left(x_{t-1}; \mu_\theta(x_t, t), \sigma_\theta(x_t, t)^2\mathbf{I}\right) \tag{2}$$

where $\mu_\theta(x_t, t)$ and $\sigma_\theta(x_t, t)$ are the mean and variance parameterized by $\theta$, respectively. Based on the literature [11], for any $\tilde{\beta}_t = \frac{1-\bar{\alpha}_{t-1}}{1-\bar{\alpha}_t}\beta_t$ $(t > 1)$ and $\tilde{\beta}_1 = \beta_1$, the parameterizations of $\mu_\theta$ and $\sigma_\theta$ are defined by:

$$\mu_\theta(x_t, t) = \frac{1}{\sqrt{\alpha_t}}\left(x_t - \frac{\beta_t}{\sqrt{1-\bar{\alpha}_t}}\epsilon_\theta(x_t, t)\right), \text{ and } \sigma_\theta(x_t, t) = \tilde{\beta}_t^{\frac{1}{2}}. \tag{3}$$

## 4 DiffTraj Framework

In this section, we present the details of the DiffTraj framework shown in Fig. 1, which applies the diffusion model to trajectory generation. The primary goal of DiffTraj is to estimate the real-world trajectory distribution $q(x_0 \mid x_0^{co})$ using a parameterized model $p_\theta(x_0^{s} \mid x_0^{co})$. Given a random noise $x_T \sim \mathcal{N}(0, \boldsymbol{I})$, DiffTraj generates a synthetic trajectory $x_0^{s}$, conditioned on the observations $x_0^{co}$. The reverse (generation) process from the Eq. (2) can be reformulated for DiffTraj as:

$$p_\theta(x_{t-1}^{s} \mid x_t^{s}, x_0^{co}) := \mathcal{N}\left(x_{t-1}^{s}; \mu_\theta(x_t^{s}, t \mid x_0^{co}), \sigma_\theta(x_t^{s}, t \mid x_0^{co})^2\mathbf{I}\right). \tag{4}$$

In this context, $\mu_\theta$ serves as a conditional denoising function that takes the conditional observation $x_0^{co}$ as input. We then extend the Eq. (3) to account for the conditional observation as follows:

$$\begin{cases} \mu_\theta(x_t^{s}, t \mid x_0^{co}) = \mu_\theta(x_t^{s}, t, \epsilon_\theta(x_t^{s}, t \mid x_0^{co})) \\ \sigma_\theta(x_t^{s}, t \mid x_0^{co}) = \sigma_\theta(x_t^{s}, t) \end{cases} \tag{5}$$

**Training**. In practice, the reverse trajectory denoising process can be summarized as learning the Gaussian noise $\epsilon_\theta(x_t^{s}, t \mid x_0^{co})$ through $x_t^{s}$, $t$ and $x_0^{co}$ (refer to Fig. 1). Then solving $\mu_\theta(x_t^{s}, t \mid x_0^{co})$ according to Eq. (3). This process can be trained by following objective [11]:

$$\min_\theta \mathcal{L}(\theta) := \min_\theta \mathbb{E}_{t, x_0, \epsilon}\left\| \epsilon - \epsilon_\theta\left(\sqrt{\bar{\alpha}_t}x_0 + \sqrt{1-\bar{\alpha}_t}\epsilon, \, t\right)\right\|^2. \tag{6}$$

The above equations show that the core of training the diffusion model is to minimize the mean squared error between the Gaussian noise $\epsilon$ and predicted noise level $\epsilon_\theta(x_t^{s}, t \mid x_0^{co})$.

**Sampling**. Given the reverse process presented in Sec. 3.2, the generative procedure can be summarized as sampling the $x_T^{s} \sim \mathcal{N}(0, \mathbf{I})$, and iterative sampling based on $x_{t-1}^{s} \sim p_\theta(x_{t-1}^{s} \mid x_t^{s}, x_0^{co})$. The final sampled output is $x_0^{s}$.

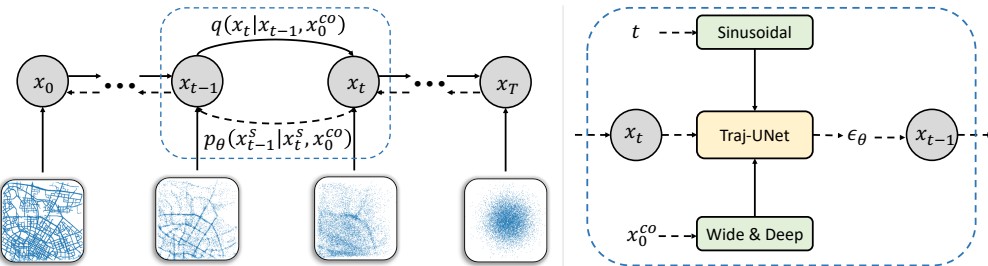

Figure 1: An illustration for trajectory generation with diffusion model. (Left) Forward and the reverse process (multiple GPS trajectories presented). (Right) Coupled the neural network model structure for reverse denoising.

## 4.1 Traj-UNet Architecture

While developing the DiffTraj framework, accurately predicting the noise level at each diffusion time step is non-trivial. A model must capture these complex spatial-temporal dependencies to accurately predict the noise level at each diffusion time step. Inaccurate noise predictions can lead to excessively noisy trajectories, resulting in unrealistic patterns. Thus, we employ Traj-UNet to predict the noise of each diffusion time step in DiffTraj, i.e., $\epsilon_\theta \left( \boldsymbol{x}_t^{\mathrm{s}}, t \mid \boldsymbol{x}_0^{\mathrm{co}} \right)$. Specifically, we construct a network based on the UNet architecture [26], which has been extensively used in generative applications involving diffusion models [35, 20, 5]. The Traj-UNet is composed of two parts: down-sampling and up-sampling, each featuring multiple 1D-CNN-based stacked residual network blocks (Resnet block) with a kernel size of 3 in each layer. An attention mechanism-based transitional module is integrated between the two parts [30] (Detailed information about Traj-UNet can be found in Fig. 4 of the **Appendix** A). To enhance the learning of noise for each diffusion time step, Traj-UNet incorporates time step embedding and then fed to each block. Specifically, we employ Sinusoidal embedding [30] to represent each $t$ as a 128-dimensional vector. Subsequently, we apply two shared-parameter fully connected layers and add them to the input of the Resnet block.

**Input trajectory**. It is important to note that the input/output data for this model is a two-dimensional trajectory tensor ($[2, \mathrm{length}]$), with each dimension representing the longitude and latitude of the trajectory, respectively. Considering CNN-based models can only accept fixed-shape data, we initially up-sample or down-sample the training trajectories using linear interpolation methods [3]. The generated trajectories are then transformed to the desired length using the same approach.

**Why use CNN based structure?** In DiffTraj, we adopt a CNN-based UNet structure over other temporal processing modules such as recurrent neural networks (RNNs) and WaveNet for several reasons. First, CNN-based methods can alleviate the vanishing gradient problem often encountered in RNNs, enabling more efficient learning of long-term dependencies. Second, compared to models like RNNs, CNNs hold a smaller number of parameters and offer greater computational efficiency, which is crucial during the reverse process. Finally, the UNet architecture excels in various generative tasks, particularly when combined with diffusion models [28, 11, 7]. Its multi-level structure with skip connections enables multi-scale feature fusion, helping the model better capture local and global contextual information in GPS trajectory data. We found that RNN and WaveNet-based architectures led to worse trajectory quality, as will be presented in Sec. 5.

## 4.2 Conditional Generation

Various external factors, such as the region of the trip and departure time, influence real-world trajectories. This conditional information should provide meaningful guidance for the generation process, ensuring that synthetic trajectories exhibit similar patterns and behaviors. In the proposed DiffTraj framework, a Wide & Deep network [6] structure is employed to effectively embed conditional information, enhancing the capabilities of the Traj-UNet model. The wide component emphasizes memorization, capturing the significance of individual features and their interactions, while the deep component concentrates on generalization, learning high-level abstractions and feature combinations. This combination effectively captures both simple and complex patterns in the trajectory. For a given trajectory, several numerical motion properties (e.g., velocity and distance)

and discrete external properties (e.g., departure time and regions) are derived. A wide network is utilized for embedding the numerical motion attributes, whereas a deep network is employed for the discrete categorical attributes. The wide network consists of a fully connected layer with 128 outputs, capturing linear relationships among motion attributes. In regard to the discrete categorical attributes, the deep network initially embeds them into multiple 128-dimensional vectors, concatenates these vectors, and subsequently employs two fully connected layers to learn intricate feature interactions. Finally, the outputs generated by the Wide & Deep networks are combined and integrated into each Resnet block.

### 4.3 Ensuring Generation Diversity

In practice, it should be avoided that conditional information results in a model with overly smooth or deterministic behavior patterns, which could undermine the intended privacy protections. To regulate the diversity of the generated trajectories and prevent the DiffTraj from following the conditional guidance too closely, we employ the classifier-free diffusion guidance method [12]. Specifically, we jointly train conditional and unconditional diffusion models with a trade-off between sample quality and diversity by using a guiding parameter $\omega$. That is, the noise prediction model can be written as:

$$\epsilon_\theta = (1 + \omega)\epsilon_\theta\left(\boldsymbol{x}_t^{\mathrm{s}}, t \mid \boldsymbol{x}_0^{\mathrm{co}}\right) - \omega\epsilon_\theta\left(\boldsymbol{x}_t^{\mathrm{s}}, t \mid \varnothing\right), \tag{7}$$

where $\varnothing$ denotes a vector in which all conditional information is $\mathbf{0}$, essentially not encoding any information. When $\omega = 0$, the model is a general diffusion model with conditional information only. By increasing the $\omega$, the model will focus more on unconditionally predicting noise (i.e., more diversity) when generating trajectories. Therefore, we can strike a balance between maintaining the quality of the generated trajectories and increasing their diversity, ensuring that the model does not generate overly deterministic behavior patterns.

### 4.4 Sampling Speed up

As described in Sec. 3.2, DiffTraj relies on a large Markov process to generate high-quality trajectories, rendering a slow reverse diffusion process. Therefore, it is a significant challenge for the proposed model to generate valid and usable trajectories in a reasonable time frame. To address this issue, we adopt a non-Markov diffusion process approach with the same optimization objective [28], thus allowing the computation of efficient inverse processes. Specifically, we can sample every $\lceil T/S \rceil$ steps with the skip-step method presented in [23]. The corresponding set of noise trajectories changes to $\{\tau_1, \ldots, \tau_S\}, \tau_i \in [1, T]$. Through this approach, the sampling steps during trajectory generation can be significantly reduced from $T$ steps to $S$ steps. Compared to the typical diffusion model [11], this method can generate higher quality samples in fewer steps [28].

### 4.5 Discussion on Privacy

The DiffTraj inherently protects privacy in GPS trajectory generation due to its generative nature and the manner in which it reconstructs synthetic data from random noise. It offers effective privacy protection through two key aspects. **(1) Generative Approach:** The DiffTraj generates trajectories by sampling from a learned distribution, rather than directly relying on or perturbing real-world data. By reconstructing trajectories from random noise during the reverse diffusion process, the model effectively decouples synthetic data from specific real data points. This ensures that the generated trajectories do not contain personally identifiable information or reveal sensitive location details, thus protecting their privacy. **(2) Learning Distribution:** The Traj-UNet learns the distribution of real trajectories and generates synthetic ones that exhibit similar patterns and behaviors, which avoids direct replicating individual real trajectories. The diverse and realistic synthetic trajectory generation ensures that sensitive information about specific individuals or their movement patterns is not inadvertently leaked. Moreover, it prevents the possibility of reverse engineering real trajectories from synthetic ones. By combining these aspects, the DiffTraj offers robust privacy protection for GPS trajectory generation, addressing concerns about data privacy while maintaining utility.

## 5 Experiments

We conduct extensive experiments on two real-world trajectory datasets to show the superior performance of the proposed DiffTraj in GPS trajectory generation. In this section, we only include

the basic setup, comparison of the main experimental results, and visualization analysis. Owing to space limitation, we leave other details (dataset, baselines, network structure, hyperparameters, etc.) and more results (conditional generation, case studies, etc.) in the **Appendix**. All experiments are implemented in PyTorch and conducted on a single NVIDIA A100 40GB GPU. The code for the implementation of DiffTraj is available for reproducibility[2].

## 5.1  Experimental Setups

**Dataset and Configuration**. We evaluate the generative performance of DiffTraj and baselines on two real-world GPS trajectory datasets of Chinese cities, namely, Chengdu and Xi'an. A detailed description of the dataset and statistics is available in **Appendix** B.1, and experiment about Porto dataset is shown in **Appendix** C.1.

**Evaluation Metrics**. To fully evaluate the quality of the generated trajectories, we follow the design in [8] and use four utility metrics at different levels. Specifically, 1) `Density error` measures the geo-distribution between the entire generated trajectory and the real one, which measures the quality and fidelity of the generated trajectories at the global level. For trajectory level, 2) `Trip error` measures the distributed differences between trip origins and endpoints, and 3) `Length error` focuses on the differences in real and synthetic trajectory lengths. Moreover, we use 4) `Pattern score` to calculate the pattern similarity of the generated trajectories (see **Appendix** B.2 for details).

**Baseline Methods**. We compare DiffTraj with a series of baselines, including three typical generative models (`VAE` [34], `TrajGAN` [9] and `DP-TrajGAN` [38]) and a range of variants based on the diffusion model approach. Specifically, `Diffwave` is a diffusion model based on the WaveNet structure [14], while `Diff-LSTM` replaces the 1D-CNN-based temporal feature extraction module with an LSTM. In addition, `Diff-scatter` and `Diff-wo/UNet` can be considered as ablation studies of the proposed method to evaluate the effectiveness of diffusion models and UNet structures, respectively. Furthermore, `DiffTraj-wo/Con` indicates no integration of the conditional information embedding module, which is used to evaluate the efficiency of the Wide & Deep module for trajectory generation. Finally, we also compare `Random Perturbation (RP)` and `Gaussian Perturbation (GP)`, two classes of non-generative approaches, to compare the superiority of generative methods in privacy preservation. The details of the implementation are presented in **Appendix** B.3.

## 5.2  Overall Generation Performance

Table 1 presents the performance comparison of DiffTraj and the selected baseline methods on two real-world datasets. Specifically, we randomly generate $3,000$ trajectories from each generative method and then calculate all metrics. From the results, we observe several trends that highlight the advantages of the proposed DiffTraj method. Non-generative methods, RP and GP, exhibit poor performance due to their reliance on simple perturbation techniques, which fail to capture the complex spatial-temporal dependencies and motion properties of the real GPS trajectories. This leads to lower-quality synthetic trajectories compared to generative models. Generative models, VAE and TrajGAN, show better performance than RP and GP but are still inferior to DiffTraj (or DiffTraj-wo/Con). This is because VAE and TrajGAN primarily focus on spatial distribution and may not adequately capture the temporal dependencies within trajectories. Additionally, the DiffTraj is designed to reconstruct the data step-by-step from random noise, which makes it more suitable for generating real GPS trajectory that reflect the inherent stochastic nature of human activities.

Diff-LSTM achieves good results in some metrics compared to the model without UNet, but falls short of DiffTraj due to the differences in the backbone network. The LSTM-based approach in Diff-LSTM is inherently designed for sequence processing and may be less effective in capturing spatial dependencies in the data compared to the CNN-based approach used in DiffTraj. The CNN-based backbone in DiffTraj can better model local spatial patterns and multi-channel dependencies more effectively, leading to higher-quality generated trajectories. Nevertheless, such results advocate the employment of diffusion models in future studies related to spatial-temporal data generation.

**Ablation Study**. The other diffusion model variants, Diffwave, Diff-scatter, and Diff-wo/UNet, do not perform as well as DiffTraj due to differences in their network structures or the absence of the UNet architecture. The Traj-UNet structure in DiffTraj is specifically designed for spatial-temporal

---

[2]https://github.com/Yasoz/DiffTraj

Table 1: Performance comparison of different generative approaches.

| Methods | Chengdu | | | | Xi'an | | | |
|---|---|---|---|---|---|---|---|---|
| | Density (↓) | Trip (↓) | Length (↓) | Pattern (↑) | Density (↓) | Trip (↓) | Length (↓) | Pattern (↑) |
| RP | 0.0698 | 0.0835 | 0.2337 | 0.493 | 0.0543 | 0.0744 | 0.2067 | 0.381 |
| GP | 0.1365 | 0.1590 | 0.1423 | 0.233 | 0.0928 | 0.1013 | 0.2164 | 0.233 |
| VAE | 0.0148 | 0.0452 | 0.0383 | 0.356 | 0.0237 | 0.0608 | 0.0497 | 0.531 |
| TrajGAN | 0.0125 | 0.0497 | 0.0388 | 0.502 | 0.0220 | 0.0512 | 0.0386 | 0.565 |
| DP-TrajGAN | 0.0117 | 0.0443 | 0.0221 | 0.706 | 0.0207 | 0.0498 | 0.0436 | 0.664 |
| Diffwave | 0.0145 | 0.0253 | 0.0315 | 0.741 | 0.0213 | 0.0343 | 0.0321 | 0.574 |
| Diff-scatter | 0.0209 | 0.0685 | – | – | 0.0693 | 0.0762 | – | – |
| Diff-wo/UNet | 0.0356 | 0.0868 | 0.0378 | 0.422 | 0.0364 | 0.0832 | 0.0396 | 0.367 |
| DiffTraj-wo/Con | 0.0072 | 0.0239 | 0.0376 | 0.643 | 0.0138 | 0.0209 | 0.0357 | 0.692 |
| Diff-LSTM | 0.0068 | 0.0199 | 0.0217 | 0.737 | 0.0142 | 0.0195 | 0.0259 | 0.706 |
| DiffTraj | **0.0055** | **0.0154** | **0.0169** | **0.823** | **0.0126** | **0.0165** | **0.0203** | **0.764** |

Bold indicates the statistically best performance (i.e., two-sided t-test with $p < 0.05$) over the best baseline.

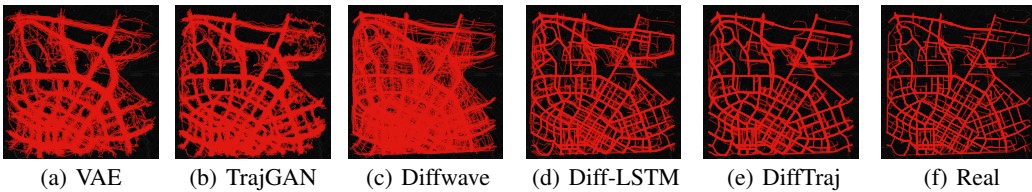

(a) VAE  (b) TrajGAN  (c) Diffwave  (d) Diff-LSTM  (e) DiffTraj  (f) Real

Figure 2: Geographic visualization of generated trajectory in Chengdu (Larger view in **Appendix** D).

modeling and multi-scale feature fusion, enabling the accurate prediction of noise levels during denoising and resulting in high-quality generated trajectory. It is also noteworthy that satisfactory results can still be accomplished when only generating scattered locations or discarding the Traj-UNet structure. Finally, the conditional version of DiffTraj outperforms the unconditional version (DiffTraj-wo/Con) because it incorporates conditional information into the trajectory generation process. This allows the model to better account for external factors such as traffic conditions, departure time, and regions, which significantly influence human mobility patterns. This result can also be verified by the performance on the pattern indicator, where this indicator significantly outperforms other models.

**Geographic Visualization**. We also visualized the generated results to better present the performance of different generative models. Fig. 2 shows the trajectory distributions generated by baseline methods for Chengdu (Please refer to Fig. 10 and Fig. 11 in **Appendix** D for Xi'an and larger view). From visualizations, we can see that all the generated trajectories show geographic profiles of the city. Notably, DiffTraj and Diff-LSTM generate trajectories almost identical to the original trajectories and significantly surpass the other methods. However, Diff-LSTM performs poorly compared to DiffTraj on sparse roads, suggesting that CNNs are skilled in capturing local and global spatial features, making them more suitable for handling sparse road scenes. In addition, developing a tailor-made UNet architecture is also essential. Comparing Fig. 2(c) and Fig. 2(e), we can observe that the latter is able to generate a visually more realistic trajectory distribution.

## 5.3  Utility of Generated Data

As the generated trajectory serves downstream tasks, its utility is critical to determine whether the data generation method is indeed practical. In this section, we evaluate the utility of generated trajectory through an inflow (outflow) prediction task, which is one of the important applications of trajectory data analysis [13, 39]. Specifically, we train multiple advance prediction models using the original and generated trajectory separately, and then compare the prediction performance (detail experimental setup refer to **Appendix** C.3). As concluded in Table 2, the performance of the models when trained on the generated data is comparable to their performance on the original data. This result shows that our generative model, DiffTraj, is able to generate trajectory that retain the key statistical properties of the original one, hence making them useful for downstream tasks such as inflow and outflow prediction. However, there is a slight increase in error metrics when using the generated data, implying there's still room for further improvements in trajectory generation. Nevertheless, this result encourages the use of generated, privacy-free alternatives to the original data in future studies related to trajectory data analysis.

Table 2: Data utility comparison by inflow/outflow prediction.

| Task | Inflow (origin/generated) | | | | Outflow (origin/generated) | | | |
|---|---|---|---|---|---|---|---|---|
| Methods | AGCRN | GWNet | DCRNN | MTGNN | AGCRN | GWNet | DCRNN | MTGNN |
| MSE | 4.33/4.55 | 4.42/4.50 | 4.45/4.62 | 4.28/4.56 | 4.45/4.70 | 4.64/4.72 | 4.72/5.01 | 4.39/4.77 |
| RMSE | 2.08/2.13 | 2.10/2.12 | 2.11/2.15 | 2.07/2.13 | 2.11/2.16 | 2.15/2.17 | 2.17/2.24 | 2.10/2.18 |
| MAE | 1.49/1.52 | 1.50/1.50 | 1.51/1.53 | 1.48/1.51 | 1.50/1.54 | 1.53/1.54 | 1.53/1.57 | 1.49/1.53 |

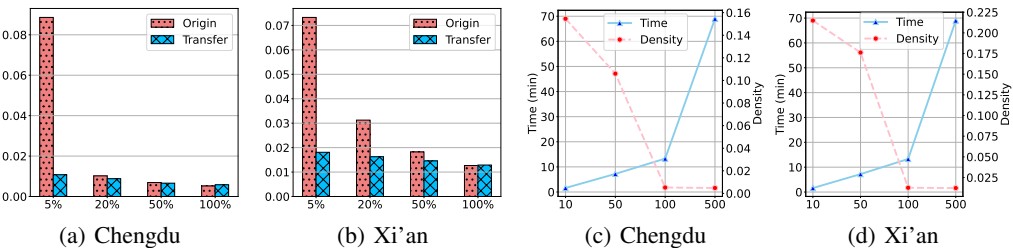

(a) Chengdu          (b) Xi'an          (c) Chengdu          (d) Xi'an

Figure 3: Additional Experiment. (a) and (b) Transfer learning; (c) and (d) Speed up sampling.

## 5.4 Additional Experiment

**Transfer Learning**. Transfer learning is an essential aspect to investigate in the context of trajectory generation, as it can demonstrate the robustness and generalization capabilities of the proposed DiffTraj framework. In this paper, we pre-train the DiffTraj model in one city, and then continue the training in another city using a different percentage ($5\% \sim 100\%$) of data. We also compared these results with only training DiffTraj on different percentages of data. As shown in Fig. 3, 'origin' indicates that transfer learning is not applicable, while 'transfer' indicates applying transfer learning. Notably, even with just $5\%$ of the data, the transfer learning model achieves a significantly lower error compared to the original one. As the percentage of data increases, the gap between the two models narrows. The results show that the DiffTraj model exhibits strong adaptability and generalization capabilities when applied to different urban scenarios. This allows the model to quickly adapt to new urban contexts and capture the inherent mobility patterns with less data, thereby reducing training time and improving applicability to diverse real-world scenarios.

**Speed up Sampling**. As discussed in Sec. 3.2, the diffusion model requires numerous sampling steps for gradual denoising, resulting in a slow trajectory generation process. By incorporating the sampling speed-up technique outlined in Sec. 4.4, we examine the efficiency of generating $50 \times 256$ trajectories using a well-trained DiffTraj on two datasets. The results are summarized in Fig. 3, where the X-axis represents the total sample steps after speed-up, and the two Y-axes indicate the time spent and Density error, respectively. All models trained with $T = 500$ diffusion steps consistently yield high-quality trajectories resembling the original distribution. Moreover, the model matches the outcomes of the no skipped steps method at $T = 100$, saving $81\%$ of the time cost. However, too few sampling steps ($S < 100$) result in a significant divergence between the generated and real trajectory. This occurs because fewer sample steps lead to more skipped reverse operations, causing Diff-Traj to discard more noise information during denoising. Nonetheless, these results highlight the substantial efficiency improvements achieved by the speed-up methods presented in Sec. 4.4.

## 6   Conclusion

In this work, we propose a new GPS trajectory generation method based on diffusion model and spatial-temporal data mining techniques. This method, named DiffTraj, leverages the data generation ability of the diffusion model and learning from spatial-temporal features by Traj-UNet. Specifically, real trajectories are gradually transformed into random noise by a forward trajectory noising process. After that, DiffTraj adopts a reverse trajectory denoising process to recover forged trajectories from the noise. Throughout the DiffTraj framework, we develop a Traj-UNet structure and incorporate Wide & Deep conditional module to extract trajectory features and estimate noise levels for the reverse process. Extensive experiments validate the effectiveness of DiffTraj and its integrated Traj-UNet. Further experiments prove that the data generated by DiffTraj can conform to the statistical properties of the real trajectory while ensuring utility, and provides transferability and privacy-preserving.

# 7 Acknowledgments

This work is supported by the Stable Support Plan Program of Shenzhen Natural Science Fund under Grant No. 20220815111111002. This research was partially supported by Research Impact Fund (No. R1015-23), APRC - CityU New Research Initiatives (No.9610565, Start-up Grant for New Faculty of City University of Hong Kong), CityU - HKIDS Early Career Research Grant (No.9360163), Hong Kong ITC Innovation and Technology Fund Midstream Research Programme for Universities Project (No.ITS/034/22MS), Hong Kong Environmental and Conservation Fund (No. 88/2022), SIRG - CityU Strategic Interdisciplinary Research Grant (No.7020046, No.7020074), Tencent (CCF-Tencent Open Fund, Tencent Rhino-Bird Focused Research Fund), Huawei (Huawei Innovation Research Program), Ant Group (CCF-Ant Research Fund, Ant Group Research Fund) and Kuaishou.

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

# A  DiffTraj Details & Hyperparameters

In this section, we cover the specific details of DiffTraj, including the DiffTraj framework, the Traj-UNet structure, and the implementation details.

## A.1  Architecture

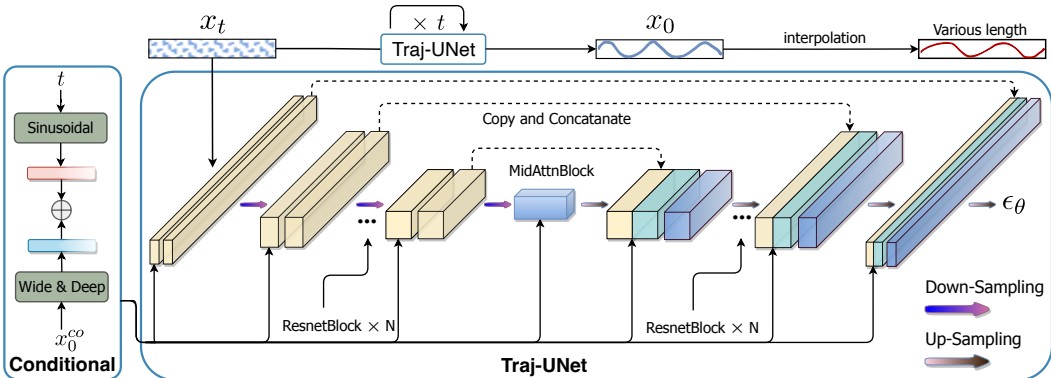

Figure 4: The network architecture used by DiffTraj in modeling $\epsilon_\theta \left( \boldsymbol{x}_t^{\text{s}}, t \mid \boldsymbol{x}_0^{\text{co}} \right)$ is divided into two modules, down-sampling and up-sampling, each containing multiple Resnet blocks.

As illustrated in Fig. 4, DiffTraj is divided into two modules, i.e., down-sampling and up-sampling, and conditional module. Each down-sampling and up-sampling module consists of multiple stacked Resnet blocks. Between the two of them, a transitional module based on the attention mechanism is integrated. To better learn the noise of each time step and guide the generation, DiffTraj integrates a conditional module to embed the time step and external traffic information, later fed to each block. Since the CNN structure can only accept data of fixed shape, we first sample each trajectory as a tensor of $[2, \text{ length}]$. Specifically, if the trajectory is below the set length, it is added using linear interpolation, and if it is greater than that, the redundant portion is removed using linear interpolation. Thus, the model will generate fixed-length trajectories, which will then be tailored to the desired length by the conditional information.

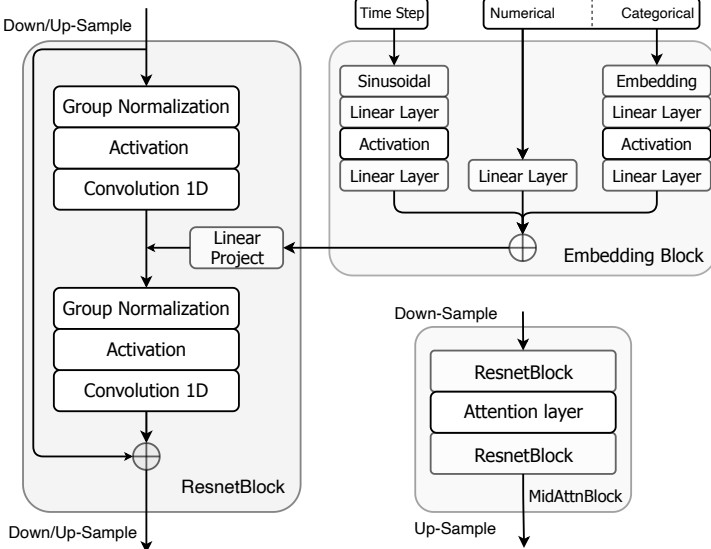

Figure 5: The main components of Traj-UNet including Resnet block, Embedding block, and middle attention block.

The main blocks of Traj-UNet are presented in Fig. 5, i.e., Resnet block, Embedding block, and middle attention block. Among them, each sampling block (down-sampling and up-sampling) consists of multiple Resnet blocks, each containing a series of group normalization, nonlinear activation, and 1D-CNN layers. Then, Traj-UNet applies up-sampling or down-sampling to the output, where down-sampling uses max pooling and up-sampling uses interpolation. After this, Traj-UNet integrates a middle attention block, which consists of two Resnet blocks and an attention layer. Note that there are no additional down-/up-sampling operations in the Resnet block. Finally, we integrate a conditional embedding block to learn the diffusion time step and conditional information. For the diffusion step, we employ Sinusoidal embedding to represent each $t$ as a 128-dimensional vector, and then apply two shared-parameter fully connected layers. For conditional information, such as distance, speed, departure time, travel time, trajectory length, and starting and ending locations, we use the Wide & Deep module for embedding. After getting the diffusion step embedding and conditional embedding, we sum them up and add them to each Resnet block.

## A.2 Implementation Details

For the proposed DiffTraj framework, we summarize the adopted hyperparameters in Table 3. In addition, to facilitate reproduction by the researcher, we provide a reference range of hyperparameters based on the experience and general settings of the present work.

Table 3: Hyperparameters setting for DiffTraj.

| Hyperparameter | Setting value | Refer range |
| --- | --- | --- |
| Diffusion Steps | 500 | $300 \sim 500$ |
| Skip steps | 5 | $1 \sim 10$ |
| Guidance scale | 3 | $0.1 \sim 10$ |
| $\beta$ (linear) | $0.0001 \sim 0.05$ | – |
| Batch size | 1024 | $\geq 256$ |
| Sampling blocks | 4 | $\geq 3$ |
| Resnet blocks | 2 | $\geq 1$ |
| Input Length | 200 | $120 \sim 200$ |

The training and sampling phase of the proposed framework is summarized in Algorithm 1 and Algorithm 2, respectively.

---
**Algorithm 1** Diffusion Training Phase
---
1: **for** $i = 1, 2, \ldots,$ **do**
2:     Sample $\boldsymbol{x}_0 \sim q(\boldsymbol{x})$,
3:     $t \sim \text{Uniform} \{1, \ldots, T\}$
4:     $\epsilon \sim \mathcal{N}(0, \mathbf{I})$
5:     $\mathcal{L} = \left\| \boldsymbol{\epsilon} - \boldsymbol{\epsilon}_\theta \left( \sqrt{\bar{\alpha}_t} \boldsymbol{x}_0 + \sqrt{1 - \bar{\alpha}_t} \boldsymbol{\epsilon}, t \mid \boldsymbol{x}_0^{\text{co}} \right) \right\|^2$
6:     $\theta = \theta - \eta \nabla_\theta \mathcal{L}$
7: **end for**
---

---
**Algorithm 2** Diffusion Sampling Phase
---
1: Sample $\boldsymbol{x}_T^{\text{s}} \sim \mathcal{N}(0, \mathbf{I})$
2: **for** $t = T, T - S, \ldots, 1$ **do**
3:     Compute $\boldsymbol{\mu}_\theta \left( \boldsymbol{x}_t^{\text{s}}, t \mid \boldsymbol{x}_0^{\text{co}} \right)$ according to Eq. (5)
4:     Compute $p_\theta \left( \boldsymbol{x}_{t-1}^{\text{s}} \mid \boldsymbol{x}_t^{\text{s}}, \boldsymbol{x}_0^{\text{co}} \right)$ according to Eq. (4)
5: **end for**
6: **return** $\boldsymbol{x}_0$
---

# B Details of the Experimental Setup

## B.1 Dataset

We evaluate the performance of DiffTraj and all baselines methods on two datasets with different cities, **Chengdu** and **Xi'an**[3]. Both datasets are collected from cab trajectory data starting from November 1, 2016, to November 30, 2016. Table 4 summarizes the statistical information of these two datasets, and Fig. 6 shows the trajectory distribution and heat map of these two datasets, where the deeper color indicates the more concentrated trajectory in the region. For all datasets, we remove all trajectories with lengths less than 120 and sample them to a set fixed length.

Table 4: Statistics of Two Real-world Trajectory Datasets.

| Dataset | Trajectory Number | Average Time | Average Distance |
|---------|-------------------|--------------|------------------|
| Chengdu | 3 493 918 | 11.42 min | 7.42 km |
| Xi'an | 2 180 348 | 12.58 min | 5.73 km |

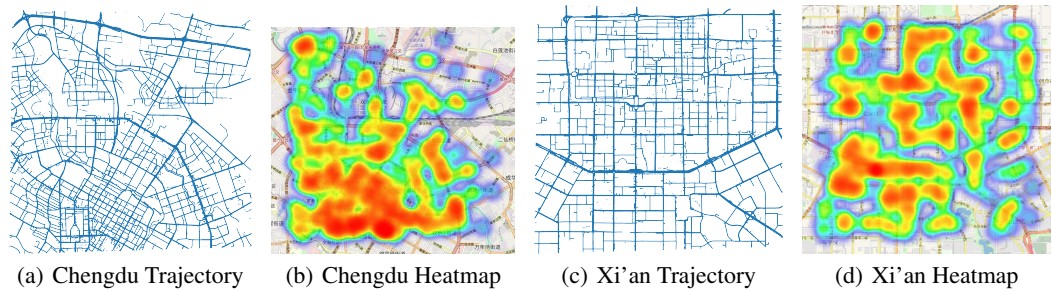

(a) Chengdu Trajectory     (b) Chengdu Heatmap     (c) Xi'an Trajectory     (d) Xi'an Heatmap

Figure 6: Origin trajectory distribution of two cities.

## B.2 Evaluation Metrics

As trajectory generation aims to generate trajectories that can replace real-world activities and further benefit downstream tasks, we need to evaluate the "similarity" between the generated trajectories and real ones. In this work, we follow the common practice in previous studies [8] and measure the quality of the generated ones by Jenson-Shannon divergence (JSD). JSD compares the distribution of the real and generated trajectories, and a lower JSD indicates a better match with the original statistical features. Suppose that the original data has a probability distribution $P$ and the generated data has a probability distribution $G$, the JSD is calculated as follows:

$$\text{JSD}(P,G) = \frac{1}{2}\mathbb{E}_P\left[\log\frac{P}{P+G}\right] + \frac{1}{2}\mathbb{E}_G\left[\log\frac{G}{G+P}\right]. \tag{8}$$

For the evaluation, we divided each city into $16 \times 16$ size grids and recorded the corresponding values for each grid. Based on this, we adopt the following metrics to evaluate the quality of the generated trajectories from four perspectives:

- **Density error:** This a global level metric that used to evaluate the geo-distribution between the entire generated trajectory $\mathcal{D}\left(\mathcal{T}_{\text{gen}}\right)$ and the real trajectory $\mathcal{D}(\mathcal{T})$.

$$\text{Density Error} = \text{JSD}\left(\mathcal{D}(\mathcal{T}), \mathcal{D}\left(\mathcal{T}_{\text{gen}}\right)\right), \tag{9}$$

  where $\mathcal{D}(\cdot)$ denotes the grid density distribution in a given trajectory set, and $\text{JSD}(\cdot)$ represents the Jenson-Shannon divergence between two distributions.

- **Trip error:** This a trajectory level metric that measures the correlation between the starting and ending points of a travel trajectory. Specifically, we calculate the probability distribution of the start/end points in the original and generated trajectories and use JSD to measure the difference between them.

---

[3]These datasets can be downloaded at https://outreach.didichuxing.com/

- **Length error:** This a trajectory level metric to evaluate the distribution of travel distances. It can be obtained by calculating the Euclidean distance between consecutive points.

- **Pattern score:** This is a semantic level metric defined as the top-$n$ grids that occur most frequently in the trajectory. We define $P$ and $P_{\text{gen}}$ to denote the original and generated pattern sets, respectively, and compute the following metrics:

$$\text{Pattern score} = 2 \times \frac{\text{Precision}\,(P, P_{\text{gen}}) \times \text{Recall}\,(P, P_{\text{gen}})}{\text{Precision}\,(P, P_{\text{gen}}) + \text{Recall}\,(P, P_{\text{gen}})} \tag{10}$$

### B.3    Baselines

In this section, we introduce the implementation of baseline methods.

- **Random Perturbation (RP):** We generate $-0.01 \sim 0.01$ random noise to perturb the original trajectory. This degree of noise ensures that the maximum distance between the perturbed points and the original does not exceed $2\,\text{km}$

- **Gaussian Perturbation (GP):** We generate Gaussian noise perturbed original trajectories with mean $0$ and variance $0.01$.

- **Variational AutoEncoder (VAE) [17, 34]:**   In this work, trajectories are first embedded as a hidden distribution through two consecutive convolutional layers and a linear layer. Then, we generate the trajectories by a decoder consisting of a linear layer and two deconvolutional layers. The size of convolution kernels in convolutional and deconvolutional layers is set to $4$ to ensure that input and output trajectories have the same size.

- **TrajGAN [9]:**   The trajectory is first combined with random noise and then passes through a generator consisting of two linear layers and two convolution layers. Subsequently, a convolutional layer and a linear layer are adopted as the discriminator. The generator and the discriminator are trained in an alternating manner.

- **Dp-TrajGAN [38]:**   The model is similar to TrajGAN, with the difference that this model replaces CNN with LSTM.In addition, we remove the unnecessary operation of differential privacy for fair comparison.

- **Diffwave [14]:**   Diffwave is a Wavenet structure model designed for sequence synthesis, which employs extensive dilated convolution. Here, we use $16$ residual connected blocks, each consisting of a bi-directional dilated convolution. Then they are summed using sigmoid and tanh activation, respectively, and fed into the 1D CNN.

- **Diff-scatter:**   We randomly sample GPS scatter points from trajectories and generate scatter points using a 4-layer MLP (neutrons: $\{128, 256, 256, 128\}$) and the diffusion model.

- **Diff-wo/UNet:**   This model uses only two Resnet blocks combined with a single attention layer between them. Compared with DiffTraj, this model does not have a UNet-type structure, which can be used to evaluate the necessity of the UNet structure.

- **Diff-LSTM:**   This model has the same UNet-type structure and number of Resnet blocks as DiffTraj, and the difference is that Diff-LSTM replaces the CNN with LSTM in Resnet block.

- **DiffTraj-wo/Con:**   DiffTraj-wo/Con represents that the Wide & Deep conditional information embedding module is removed. The rest is the same as DiffTraj.

## C    Additional Experiments

### C.1    Porto Dataset Experiment

To further validate the generalizability of our approach, we conducted experiments on a different **Porto** dataset. Through the new experiment, we can see that DiffTraj achieves the best performance on this dataset, demonstrating its ability to adapt to different geographic regions and capture different trajectory patterns. These results underscore the generalizability across different cities.

Table 5: Performance comparison of different generative approaches on Porto dataset.

| Methods | Density ($\downarrow$) | Trip ($\downarrow$) | Length ($\downarrow$) | Pattern ($\uparrow$) |
|---------|---------|---------|---------|---------|
| VAE | 0.0121 | 0.0224 | 0.0382 | 0.763 |
| TrajGAN | 0.0101 | 0.0268 | 0.0332 | 0.765 |
| Diffwave | 0.0106 | 0.0193 | 0.0266 | 0.799 |
| Diff-LSTM | 0.0092 | 0.0141 | 0.0255 | 0.828 |
| DiffTraj | 0.0087 | 0.0132 | 0.0242 | 0.847 |

## C.2 Data Utility Additional Experiments

To illustrate the performance of DiffTraj generated trajectories, we use the trajectory data generated based on different baselines for the inflow/outflow prediction task. As can be seen from the results in Table 6, the training results using DiffTraj to generate trajectories are closer to the real environment than using the baseline generation method, which is in line with the trend shown in Table 1.

Table 6: Data utility comparison by inflow/outflow prediction (different baselines).

| Task | Inflow | | | | Outflow | | | |
|------|--------|--------|----------|----------|---------|--------|----------|----------|
| Baseline | GAN | VAE | Diffwave | DiffTraj | GAN | VAE | Diffwave | DiffTraj |
| MSE | 6.78 | 6.79 | 4.62 | 4.50 | 5.97 | 5.81 | 4.75 | 4.72 |
| RMSE | 2.60 | 5.61 | 2.16 | 2.12 | 2.44 | 2.41 | 2.18 | 2.17 |
| MAE | 1.76 | 1.76 | 1.52 | 1.50 | 1.68 | 1.66 | 1.56 | 1.54 |

## C.3 Data Utility Setup

In this paper, we use inflow/outflow traffic forecasting to test the utility of the generated data. Inflow/outflow traffic forecasting is a critical task in urban traffic management that involves predicting the volume of vehicles entering (inflow) or leaving (outflow) a specific region within a certain period of time. In this experiment, we divided a city into $16 \times 16$ grids, where each grid represents a specific region. The traffic volume entering (inflow) or leaving (outflow) each region within a certain period is predicted. The primary goal of this experiment is to train various prediction models using both original and generated trajectory data, comparing their prediction performance. This evaluation provides an important perspective on the real-world applicability of the data generated by DiffTraj, assessing not just the fidelity of the generated trajectories, but also their utility in downstream tasks. In the experimental setup, we train the prediction models using the generated data and the original data separately, and then test their prediction performance on real data. Advanced neural network models, such as AGCRN, Graph WaveNet, DCRNN, and MTGNN, have been employed to handle this task due to their ability to capture complex spatial-temporal dependencies in multivariate time series data. All the above models and code in this section are followed the publicly available code[4] provided in the literature [13].

- **AGCRN (Adaptive Graph Convolutional Recurrent Network):** AGCRN is a sophisticated model for spatial-temporal forecasting, which leverages both spatial and temporal features of data. It uses graph convolution to capture spatial dependencies and RNNs to model temporal dynamics, making it capable of handling complex spatial-temporal sequences.

- **GWNet (Graph WaveNet):** GWNet is designed for high-dimensional, structured sequence data. It incorporates a Graph Convolution Network (GCN) to model spatial correlations and a WaveNet-like architecture to model temporal dependencies. The combination allows for capturing both the spatial and temporal complexities present in high-dimensional data.

- **DCRNN (Diffusion Convolutional Recurrent Neural Network):** DCRNN is a deep learning model designed for traffic forecasting, which handles the spatial and temporal dependencies in traffic flow data. It uses a diffusion convolution operation to model spatial dependencies and a sequence-to-sequence architecture with scheduled sampling and residual connections to model temporal dependencies.

---

[4]https://github.com/deepkashiwa20/DL-Traff-Graph

- **MTGNN (Multivariate Time-series Graph Neural Network):** MTGNN is a model that captures complex spatial-temporal relationships in multivariate time series data. The model leverages a graph neural network to model spatial dependencies and an auto-regressive process to capture temporal dependencies. It also uses a gating mechanism to adaptively select the relevant spatial-temporal components, thus improving the forecasting performance.

For accuracy comparison, we use the mean square error (MSE), root mean square error (RMSE), and mean absolute error (MAE) as metrics to assess the prediction accuracy of all methods. These three metrics are defined as follows:

$$\text{MSE}(\mathbf{X}, \ \hat{\mathbf{X}}) = \frac{1}{N} \sum_{i}^{N} \left( X^{(i)} - \hat{X}^{(i)} \right)^2, \tag{11}$$

$$\text{RMSE}(\mathbf{X}, \ \hat{\mathbf{X}}) = \sqrt{\frac{1}{N} \sum_{i}^{N} \left( X^{(i)} - \hat{X}^{(i)} \right)^2}, \tag{12}$$

$$\text{MAE}(\mathbf{X}, \ \hat{\mathbf{X}}) = \frac{1}{N} \sum_{i}^{N} \left| X^{(i)} - \hat{X}^{(i)} \right|, \tag{13}$$

where $X^{(i)}$ and $\hat{X}^{(i)}$ are the ground truth and predicted inflow/outflow value at time $i$, respectively.

### C.4 Conditional Generation

As we described in Sec. 4.2 and Sec. 4.3, the DiffTraj framework takes into account various external factors that influence real-world trajectories, such as road network structure and departure time. These factors are used to guide the generation process, ensuring the synthetic trajectories mimic real-world patterns and behaviors. The model employs a Wide & Deep network to effectively embed this conditional information, enhancing the capabilities of the Traj-UNet model.

To evaluate the conditional generation capability of DiffTraj, we investigate the case of generated trajectories where the start and end regions of the trajectories were pre-defined. The model was tasked to generate 20 random trajectories adhering to these conditions. The results depicted in Fig. 7 and Fig. 8 (The red and blue boxes indicate the starting and ending regions, respectively), effectively demonstrate proficiency in generating trajectories that align with the specified start and end regions of DiffTraj. This is observed consistently across both cities under study, reinforcing the model's ability to accurately incorporate and adhere to conditional information. This robust capability underscores the versatility of DiffTraj in generating meaningful trajectories under specific conditions, and its applicability in real-world scenarios where such conditions often exist.

### C.5 Ensuring Generation Diversity

In addition, DiffTraj is designed to generate high-quality trajectories and ensure a level of diversity that prevents overly deterministic behavior patterns, thereby upholding intended privacy protections. By integrating a classifier-free diffusion guidance method, DiffTraj can strike a calculated balance between sample quality and diversity. To validate the capacity for generating diverse trajectories of DiffTraj, we devised an experiment that manipulates the guiding parameter, $\omega$. This experiment aimed to examine the quality-diversity balance in trajectories generated by DiffTraj, and how this equilibrium responds to variations in $\omega$. In this experimental setup, we studied the trajectories yielded under different $\omega$ settings (specifically $\omega \in 0.1, 1, 10$) while keeping the conditional information the same.

The results, as illustrated in Fig. 7 and Fig. 8, reveal an unambiguous link between an increase in $\omega$ and a rise in trajectory diversity. This finding affirms that DiffTraj adeptly manages the balance between diversity and quality. As $\omega$ increases, the model demonstrates a tendency to spawn more varied trajectories. This is because a higher $\omega$ prompts the model to place more emphasis on unconditional noise prediction and reduce the sway of the conditional information. Thus, the model grows more proficient at creating diverse trajectories, albeit potentially compromising some quality. This greater diversity is a consequence of the model having fewer constraints from specific conditions, providing more latitude to explore a wider range of possible trajectories. This experiment underscores

the flexibility and control inherent in DiffTraj in balancing trajectory quality and diversity, vital characteristics for generating realistic and diverse trajectories. Therefore, $\omega$ serves as a control knob for modulating the trade-off between trajectory quality and diversity, providing a powerful tool for users to align the generated trajectories with specific application requirements.

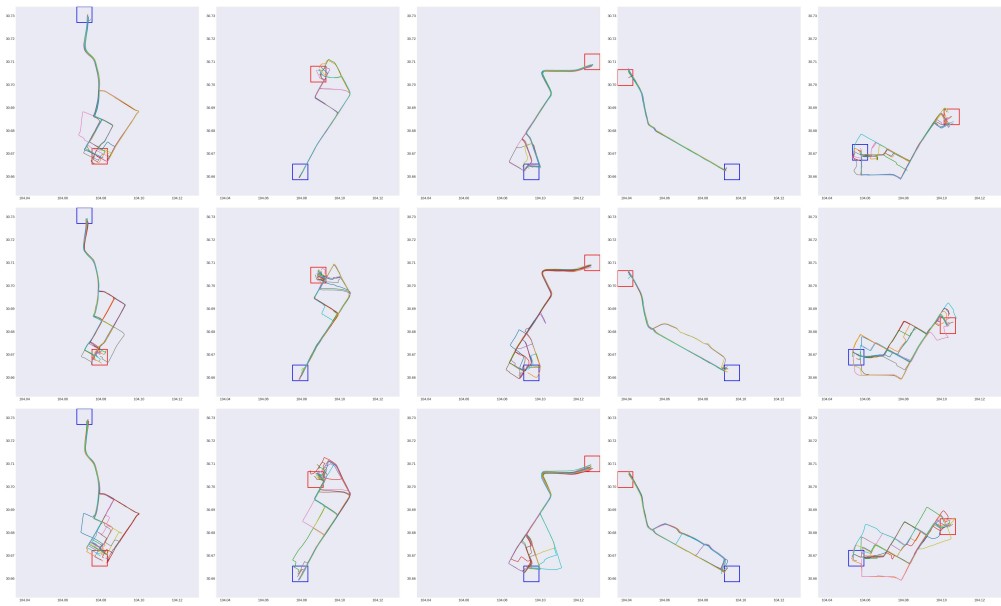

Figure 7: Conditional trajectory generation on Chengdu. The guidance scales $\omega$ of the first, second and third rows are $0.1, 1, 10$, respectively. The rectangular box indicates the area of the assigned start and end points.

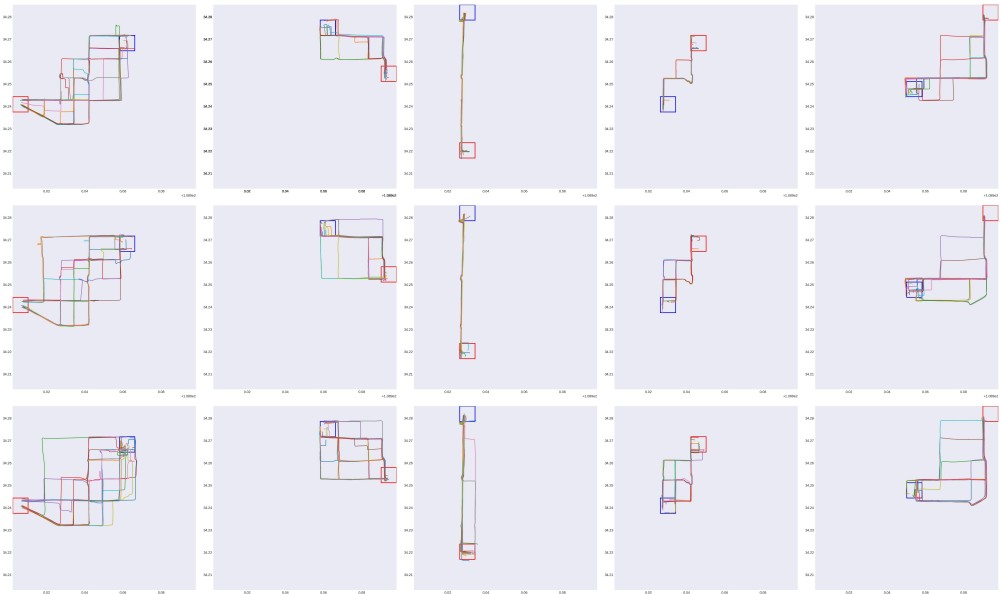

Figure 8: Conditional trajectory generation on Xi'an. The guidance scales $\omega$ of the first, second and third rows are $0.1, 1, 10$, respectively. The rectangular box indicates the area of the assigned start and end regions.

# D  Visualization Results

We append a series of experimental results in this section due to space constraints. As shown in Fig. 9, we visualize the heat map of the trajectory distribution with multiple resolutions. Specifically, we divide the whole city into $32 \times 32$, $16 \times 16$, and $8 \times 8$ grids, and then count the distribution of trajectories in each grid. The comparison clearly indicates that the distributions are highly consistent from all resolutions. The visualized results also verify the effectiveness of metrics in Table 1, revealing that the proposed model can generate high-quality trajectories with remarkable accuracy and retain the original distribution.

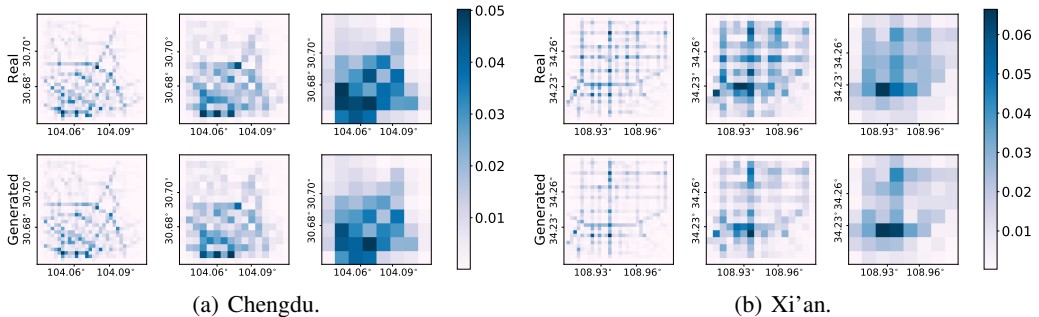

(a) Chengdu.  (b) Xi'an.

Figure 9: Comparison of the real and generated trajectory distributions with different resolutions. The city is divided into different size grids ($32 \times 32$, $16 \times 16$, and $8 \times 8$ grids).

In addition, we also show the geographic results of the trajectories generated by different generation methods for two cities, Chengdu and Xi'an. The visualization results are concluded in Fig. 10 and Fig. 11.

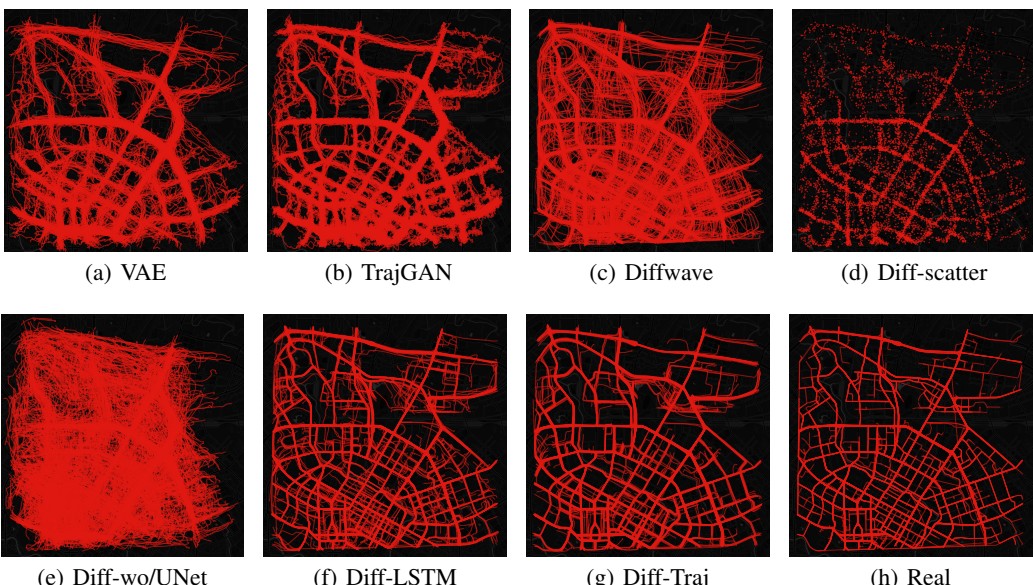

(a) VAE  (b) TrajGAN  (c) Diffwave  (d) Diff-scatter

(e) Diff-wo/UNet  (f) Diff-LSTM  (g) Diff-Traj  (h) Real

Figure 10: Geographic visualization of generated trajectory in Chengdu.

The rest visualize the forward trajectory addition noise process and reverse trajectory denoising process of Diff-Traj.

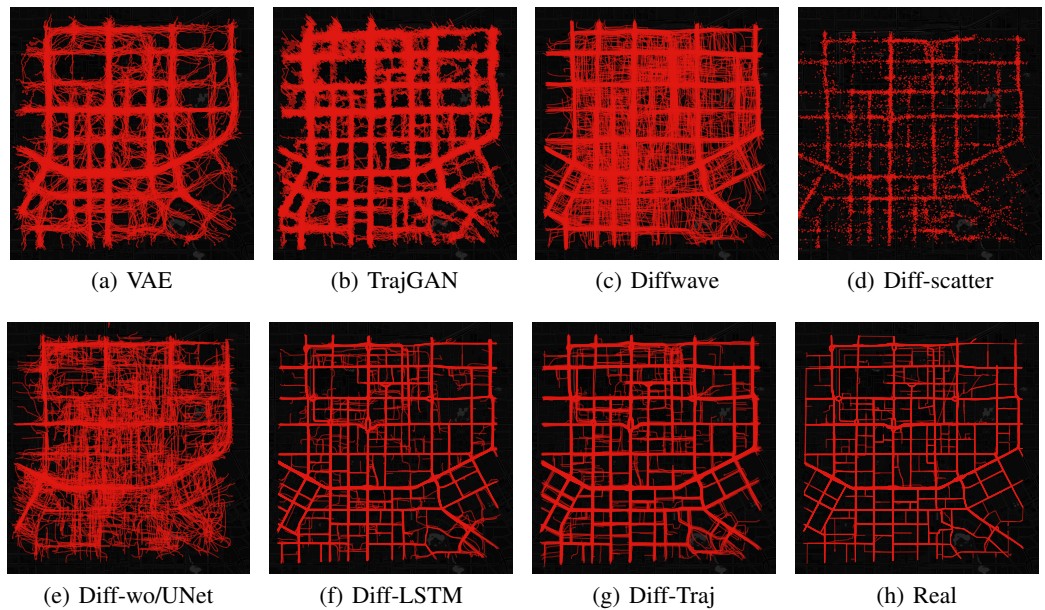

Figure 11: Geographic visualization of generated trajectory in Xi'an.

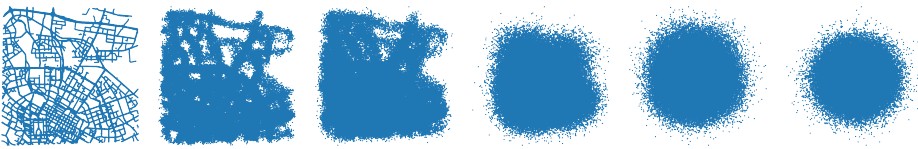

Figure 12: Forward trajectory noising process (Chengdu).

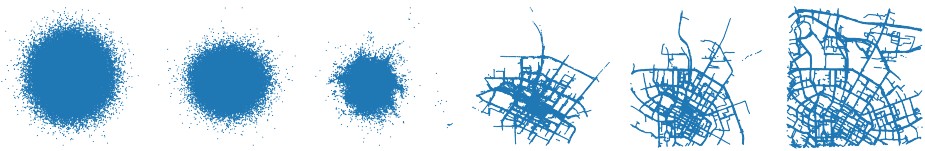

Figure 13: Reverse trajectory denoising process (Chengdu).

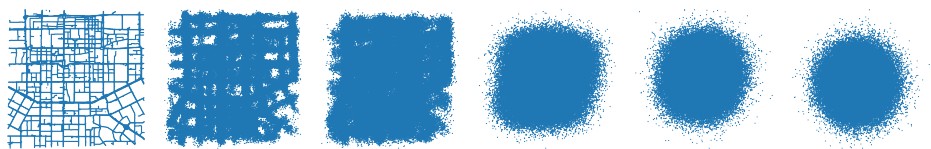

Figure 14: Forward trajectory noising process (Xi'an).

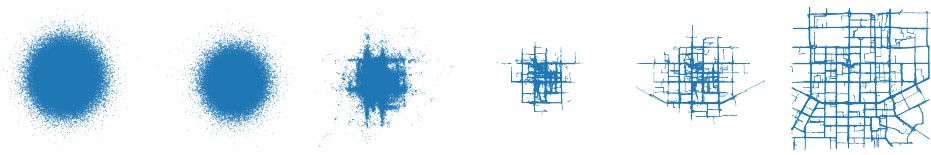

Figure 15: Reverse trajectory denoising process (Xi'an).

