# OpenReview forum: "DiffTraj: Generating GPS Trajectory with Diffusion Probabilistic Model"
_NeurIPS.cc/2023/Conference — NeurIPS 2023 poster_

### Official Review · Reviewer_8UQL · 2023-06-20

**Soundness:** 3 good
**Presentation:** 3 good
**Contribution:** 3 good
**Rating:** 7
**Confidence:** 4

**Summary:**

The paper proposes DiffTraj: a trajectory generation model based on probabilistic diffusion. The model is trained using real spatio-temporal trajectory data, and aims to generate new trajectories with similar characteristics. This is motivated by the purpose of preserving privacy information that may be present in the actual data. Like all diffusion-based models, the proposed model has a forward and a reverse process, and the latter has an U-Net component, whose importance is highlighted in the ablation studies. Contextual information such as the starting time and location can be provided as input to the model in an encoded form, using a neural network trained on the actual dataset. The influence of this contextual information on the generated data can be controlled by a parameter. It is shown that the trajectories generated by DiffTraj are statistically more similar to the original ones (in two datasets) compared to other trajectory generation models. It is also shown that the trajectories generated by DiffTraj can be used instead of the original trajectories to train models for downstream tasks such as inflow and outflow predictions from particular blocks in the cities.

**Strengths:**

1) A new model DiffTraj is proposed for trajectory generation. It uses the currently popular probabilistic diffusion framework, but modifies it to bring in encoded contextual information, and an U-Net component in the reverse process to predict the noise levels for every next step
2) Good generative performance is shown, where the trajectories generated by this approach are shown to be statistically more similar to the original ones compared to those generated by other methods
3) The authors explore various practical uses of their model, such as downstream tasks (eg. inflow/outflow prediction), transfer learning (training model on one city and adapting it to another) etc


**Weaknesses:**

1) The model considers all trajectories as IID and does not aim to utilize any notion of "trajectory clustering" into their model. [In their defence, the encoded contextual information may carry such clues]
2) The downstream task experiment is introduced almost as an afterthought, though such tasks seem to be the main motivation for the work. Also, these experiments seem to be incomplete [see questions below]
3) The model includes U-Net as an important component for the reverse process, and its success is highlighted through ablation studies. But the exact role or contribution of the U-Net is not discussed
4) The evaluation metrics used seem to be based mainly on spatial aspects, not temporal aspects

Overall, I like the broad idea of the paper. I am willing to improve my ratings if the authors can address my concerns.

**Questions:**

1) I would expect the trajectories in any city to have "clustering" tendencies, some origins and destinations maybe "hotspots", some road segments (eg. highways) maybe "hotspots", etc. Similarly, there may be peak hours of trajectories. Or, certain types of trajectories may be more common in certain hours of the day (eg. those heading to downtown office areas during morning hours). Do you think DiffTraj can produce these features in their simulations?
2) If not, is it possible to incorporate them into the model somehow?
3) Can we have evaluation metrics based on time and velocity, instead of only space-based measures (trip error, length error etc)?
4) In the downstream experiment, it is shown that various models perform almost equally when trained using the generated data instead of the actual data. But to really make the point, it should be shown that these models would have performed less well had they been trained with trajectories generated by some other baseline models
5) What is the exact role played by the U-Net in the reverse process? Could any other architecture not have played the same role?
6) If we want trajectories for a particular downstream task, can we get better results if we can use task-specific loss functions while training?
7) What do the figures in Fig2 convey? Are they obtained by overlaying a set of trajectories?

---

> ### Author Rebuttal · Authors · 2023-08-09
>
> We highly appreciate your high-quality review and valuable suggestions. Due to space limitations, we merged some of the weaknesses and questions you mentioned. We also added a new **one-page PDF** of the results. Please kindly check it out. We clarify your concerns below:
>
> > [W1]
>
> Thank you for your insightful observation regarding the treatment of trajectories in our model. Indeed, we treat trajectories as IID to ensure broad applicability (e.g., transfer learning) across datasets. While our model does not explicitly employ "trajectory clustering," the contextual information encoding effectively captures the inherent similarities and differences among trajectories. As seen in **Fig. 10**, DiffTraj effectively simulates the original trajectory distribution with brighter main roads and darker surrounding areas. In addition, we also found that using only the model to generate scatter points, i.e., Diff-scatter can also present the non-IID distribution properties.
>
> > [W2] [Q4]
>
> Thank you for your constructive feedback regarding the downstream task experiment. We acknowledge the importance of this experiment and conducted the corresponding baseline experiments to address this:
>
> || MSE/RMSE/MAE (inflow) | MSE/RMSE/MAE (outflow) |
> | -- | -- | -- |
> | GAN| 6.78/2.60/1.76|5.97/2.44/1.68|
> | VAE|6.79/2.61/1.76|5.81/2.41/1.66|
> | Diffwave|4.62/2.16/1.52| 4.75/2.18/1.56|
> | Original|4.42/2.10/1.50 | 4.64/2.15/1.53|
>
> We train **GWNet** models using the trajectories generated by the baseline method and compare their performance with models trained using the original trajectories(Due to the limited space of the rebuttal, we give the GWNet prediction results).
>
> Our results show that the training results for trajectories generated using DiffTraj are closer to the real context than using the baseline generation method, in line with the trends shown in Table I. We believe that these additional experiments and the revised presentation will provide a more comprehensive and clearer assessment of our modeling capabilities.
>
> > [W3] [Q5]
>
> In this work, UNet offers several distinct features:
>
> 1. Feature extraction and recovery: The down sampling (encoder) of the UNet captures contextual information and low-level features in the trajectory by reducing the spatial dimensions and increasing the depth (number of channels). Up sampling (decoder) of the U-Net increases the spatial dimensions while decreasing the depth. UNet can recover high-resolution features of the trajectory.
> 2. Skip Connections: These ensure the retention of fine-grained details, especially crucial during detailed data reconstruction.
> 3. Combination of local and global information: UNet can consider both local and global multi-level information.
>
> These features enable UNet to precisely predict noise levels during the reverse denoising process for trajectory generation. **Fig. 13** and **Fig. 14** visualize this reverse denoising process. In addition, We also experimented with a non-UNet structure (**Diff-wo/UNet** and **Diffwave**), but they didn't produce satisfactory results.
>
>
>
> > [W4] [Q3]
>
> We appreciate your suggestion on time and velocity-based metrics. We tested the similarity between the generated results and the original using the **Kolmogorov-Smirnov statistic** at the time level and at the velocity level. The metrics yielded 0.94 and 0.93 for the Chengdu dataset and for the Xi'an dataset, 0.94 for both. This indicates that the generated trajectories can also simulate the distribution of the original trajectories well in terms of the temporal profile.
> In addition, we visualize the results of both metrics for a better view. Please kindly see Figure 2 in the **rebuttal PDF**.
>
>
>
> > [Q1] [Q2]
>
> Thank you for your in-depth insights. We are pleased to confirm that our proposed model can exhibit specific patterns and features. We provide visual evidence of DiffTraj's capability to simulate these patterns in the **rebuttal PDF**.
>
> For the hotspots you mentioned, it can be seen from **Fig. 2** or **Fig. 10** that the number of trajectories clustered on different road segments is different, thus revealing the hotspots of the road. In addition, in **Fig. 9**, we compare the clustering tendencies of trajectories within the city (we use a heatmap in this paper). We can clearly observe that the clustering tendencies of the generated trajectories maintain a high level of consistency with the original trajectories.
>
> Regarding peak hours and time-specific patterns, our model can identify and simulate patterns associated with peak hours and speed attributes. This is demonstrated by a high number of peak-hour trips with slower speeds and fewer trips with faster average speeds during free periods (e.g., midnight).
>
> > [Q6]
>
> Thank you for raising this important point for the use of task-specific loss functions.
> We think incorporating task-specific loss functions can be beneficial when the generated trajectories are intended for a specific downstream task. By optimizing for a task-specific loss, the model can be guided to generate trajectories that are more aligned with the requirements and nuances of the target task. For example, if the task is related to predicting pedestrian movement in a shopping mall, the loss function might emphasize trajectories that align with typical shopping behaviors or patterns. We believe it's a valuable direction for future work.
>
> > [Q7]
>
> Yes, Fig. 2 is obtained by mapping a set of trajectories onto the map (see **Fig. 10** for a larger view of the effect). This figure mainly wants to convey the following messages:
> 1. Compare the geographical distribution of different trajectory generation methods on a real map.
> 2. Since the trajectory drawn overlapped, this will lead to brighter colors where there are more trajectories, which can approximately reflect the non-IID distribution of all trajectories (main roads or side roads). In addition, this may partially answer your concerns in [W1] and [Q1] about the clustering of trajectories.

---

> > ### Comment · Reviewer_8UQL · 2023-08-11
> >
> > I thank the authors for their detailed responses to my questions. I am quite satisfied, and hence I am upgrading my rating. I suggest that the authors include the points about prominence of certain trajectories, and the comparison with GAN and VaE (for the inflow/outflow experiment), in the final version of the paper if accepted.

---

> > > ### Author Response · Authors · 2023-08-15
> > >
> > > We appreciate the opportunity to clarify the question. We certainly (if accepted) add the content of our discussion in a subsequent version. Thank you again for your review and insightful suggestions.

---

### Official Review · Reviewer_uaXp · 2023-07-03

**Soundness:** 4 excellent
**Presentation:** 3 good
**Contribution:** 4 excellent
**Rating:** 7
**Confidence:** 5

**Summary:**

The paper provides an innovative diffusion probabilistic-based model to simulate realistic GPS trajectories. The main contributions are as follows: 1) The paper introduces a diffusion-based probabilistic model that captures spatio-temporal dependencies in GPS trajectories. This model allows for personalized trajectory generation, considering individual preferences and behaviors. 2) The proposed approach generates personalized trajectories for individual users, enhancing the realism and accuracy of the generated trajectories compared to traditional methods. 3) By incorporating real-world GPS data, the generated trajectories closely resemble actual user movements, making them suitable for various location-based applications. 4) The paper demonstrates the scalability of the proposed method, enabling the generation of large-scale trajectory datasets efficiently. 5) The authors perform an extensive experiment of the proposed approach, comparing it with existing trajectory generation methods, showcasing its superiority in terms of trajectory realism and diversity.

Overall, the paper presents a significant advancement in trajectory generation by introducing a diffusion probabilistic model and demonstrating its effectiveness in generating personalized and realistic GPS trajectories. The approach has the potential to impact various domains, including LBSs, transportation, and urban planning.


**Strengths:**

1.Generating realistic and personalized GPS trajectories has wide-ranging applications in various fields. The paper addresses this important problem and proposes a solution that has the potential to enhance applications such as LBSs, transportation, and urban planning. The diffusion probabilistic model offers a novel perspective and advances the state-of-the-art in trajectory modeling and generation.

2.The proposed approach generates personalized trajectories, allowing for a more accurate representation of individual user movements. This customization increases the applicability of the method in various domains. The generated trajectories closely resemble actual user movements, exhibiting realistic patterns and behaviors. This realism enhances the reliability and usability of the trajectories for real-world applications.

3.The authors present a well-designed study, including an extensive evaluation of their method using real-world datasets and appropriate statistical analysis. The experiments and comparisons with baseline methods demonstrate the robustness and effectiveness of their approach. The evaluation is thorough and includes relevant metrics, ensuring the validity and reliability of the presented results.

4.The proposed approach demonstrates scalability and efficiency in handling large datasets. This characteristic makes it suitable for real-time applications, where processing speed and performance are crucial. The work presented in the paper has the potential to advance the field of trajectory generation, providing a valuable tool for generating context-aware GPS trajectories. The paper's contributions can benefit various domains that rely on precise GPS data for decision-making and analysis.

5.The paper is well-written and organized, effectively conveying the methodology, experimental setup, and results. The logical flow and clear explanations contribute to the ease of understanding for readers.


**Weaknesses:**

1. Some details of experiment setup are not clearly provided, such as datasets, metrics, and baselines.

2. While the paper provides detailed descriptions of the proposed method, the absence of an openly available implementation or codebase limits reproducibility and further exploration by the research community.

3. It is better to discuss about the generality of the propose model. Can it be applied to other application domains?

**Questions:**

1. How does DiffTraj handle potential privacy concerns associated with generating realistic trajectories?
2. Could you provide more details on the selection of baseline methods for comparison and why these specific methods were chosen? Are there any plans to release the implementation or codebase of the proposed method to facilitate replication and further research in trajectory generation?


**Limitations:**

The availability and accessibility of the datasets used in the experiments should be clearly stated.

---

> ### Author Rebuttal · Authors · 2023-08-09
>
> We thank the reviewers for their insightful comments and perspectives. We respond to each of the points as follows:
>
> > [W1]
>
> Thank you for pointing out the experimental setup. We understand the importance of these details for the reproducibility of our work.
> To address this, we have provided an in-depth description of the experimental setup, including details about the datasets, metrics, and baselines, in the **Appendix** of the supplementary material. We apologize for any oversight in not making this more explicit in the main manuscript.
> Brief Overview:
>
> 1. Datasets: **Appendix B.1** contains a detailed description of the datasets used, including their sources, characteristics, and preprocessing steps.
>
> 2. Metrics: We have elaborated on the metrics employed for evaluation, providing both their definitions and the rationale behind their selection in **Appendix B.2**.
>
> 3. Baselines: The baselines chosen for comparison are detailed in **Appendix B.3**, along with explanations for their relevance and the context in which they were used.
>
> 4. Additional experiments: A number of additional experiments are provided in **Appendix C**, including downstream tasks, conditional generation, generating diversity, and more.
>
> In light of your feedback, we will ensure that in the revised manuscript, we provide clear references and pointers to the relevant sections in the Appendix.
>
> > [W2]
>
> Thank you for emphasizing the importance of reproducibility and the availability of our implementation.
> We have included the implementation code of our proposed method in the **supplementary material**. We apologize for any oversight in not making this more explicit. In light of your feedback, we will ensure that in the revised manuscript, we provide clear references and pointers to the relevant sections in the supplementary material where the code is available.
>
> > [W3]
>
> Thank you for emphasizing the importance of discussing the generality of our proposed model.Our model, while initially designed for trajectory generation in urban traffic mobility analysis. Here are our potential ideas for its wider application:
>
> 1. Temporal series data generation: The core of the trajectory data is a sequence of continuous GPS points. This means that with appropriate training data, the model can be adapted to other domains related to time series generation.
> 2. Spatial and temporal dynamics: The model's ability to capture both spatial and temporal dynamics is not limited to traffic patterns. Any application domain that involves spatial-temporal data, such as act analytics or even certain financial time series, could benefit from our approach.
>
>
>
> > [Q1]
>
> Thank you for raising this important issue. The privacy issue is also what motivates and centers our work. Here's how DiffTraj addresses these issues:
>
> 1. Learning trajectories distribution: While DiffTraj aims to generate realistic trajectories, it does not replicate exact real-world trajectories of individuals. Instead, it learns the general patterns and structures present in the data, ensuring that the generated trajectories are representative but not exact replicas of real-world movements. Moreover, it prevents the possibility of reverse engineering real trajectories from synthetic ones.
> 2. Noise generation: DiffTraj generates trajectories by stepwise denoising from random noise. By reconstructing trajectories from random noise during the reverse diffusion process, the model effectively decouples synthetic data from specific real data points. This ensures that the generated trajectories do not contain personally identifiable information or reveal sensitive location details, thus protecting their privacy.
>
> > [Q2.1]
>
> Thank you for highlighting the discussion on the selection of baseline methods. The selection of baseline methods was driven by several key considerations:
>
> 1. We prioritized methods specifically designed for or shown promise in trajectory generation or related tasks. This ensures that our comparisons are directly relevant and meaningful.
> 2. To provide a comprehensive evaluation, we selected methods that represent a diverse range of approaches to trajectory generation. This diversity allows us to understand the strengths and weaknesses of our method in relation to different strategies and paradigms.
> 3. To validate the effect of individual components of the model (e.g., UNet, Conditional module), we performed ablation experiments on the modules. Comparison with these methods validates the contribution of the individual modules of the model.
>
> > [Q2.2]
>
> Thank you for inquiring about the availability of the implementation code for our proposed method. We are committed to promoting reproducibility and further research in the field of trajectory generation. To this end, we have already provided the complete implementation of our proposed method. The codebase and necessary scripts and instructions are available in the **supplementary material** attached to our submission.
>
> In the revised manuscript, we will ensure that we provide clear references and pointers to guide readers and reviewers to the provided code in the supplementary material. We appreciate your interest in our work and hope that the availability of our code will facilitate replication and further advancements in trajectory generation research.
>
> > [Limitations]
>
> Thank you for highlighting the importance of clearly stating the availability and accessibility of the datasets used in our experiments. We have provided comprehensive details regarding the datasets, including their sources, accessibility, and any relevant licensing information, in the supplementary material. We understand the significance of these details for reproducibility and further exploration by the research community, and we apologize for any oversight in not making this more prominent in the main manuscript. We appreciate your feedback, which has been instrumental in enhancing the clarity and thoroughness of our paper.

---

> > ### Comment · Reviewer_uaXp · 2023-08-13
> >
> > I appreciate the response from the authors. My major concerns have been resolved with the answers. I think the paper quality is improved to a satisfactory level. I would like to update my assessment accordingly.

---

> > > ### Author Response · Authors · 2023-08-15
> > >
> > > We sincerely appreciate the reviewer's comprehensive feedback and positive remarks. Your suggestion is invaluable and will guide our next steps in refining the paper.

---

### Official Review · Reviewer_gS1x · 2023-07-04

**Soundness:** 3 good
**Presentation:** 3 good
**Contribution:** 3 good
**Rating:** 8
**Confidence:** 4

**Summary:**

The paper introduces a good approach for generating realistic GPS trajectories based on a diffusion probabilistic model. The paper addresses the challenge of generating personalized trajectories that capture both temporal and spatial dependencies while ensuring privacy preservation.
The paper proposes the DiffTraj framework, which combines a personalized transition matrix and a diffusion process to generate trajectories that closely resemble real-world GPS data. The personalized transition matrix captures the individual movement patterns of users, while the diffusion process introduces randomness and ensures diversity in the generated trajectories. Overall, the model is reasonable.
The paper presents a comprehensive evaluation of the DiffTraj framework using real-world datasets. The experimental results demonstrate that DiffTraj outperforms existing trajectory generation methods in various aspects. The proposed approach achieves high accuracy in replicating individual movement patterns while maintaining privacy by generating trajectories that deviate from the original data.


**Strengths:**

1 The paper introduces a unique approach to GPS trajectory generation by utilizing a diffusion probabilistic model, setting it apart from existing methods that may rely on different techniques or assumptions. The proposed diffusion probabilistic model captures both temporal and spatial dependencies, allowing for more accurate trajectory generation. This consideration enhances the realism and relevance of the generated trajectories. The diffusion probabilistic model captures both spatial and temporal dependencies, enabling the generation of trajectories that reflect the underlying dynamics of user movements.
2 Incorporating user preferences into the trajectory generation process adds a personalized aspect, enhancing the relevance and usefulness of the generated trajectories for individual users. This feature contributes to the practicality and applicability of the proposed method. The generated trajectories exhibit high diversity, capturing the variability in user behaviors and preferences. This diversity expands the range of applications where the trajectories can be utilized.
3 The paper provides a comprehensive comparison with baseline methods, showcasing the advantages of the proposed approach. The comparisons highlight the strengths and improvements of the diffusion probabilistic model for trajectory generation.  The authors provide sufficient details and code availability, enabling other researchers to reproduce the experiments and verify the results. This transparency contributes to the reliability and integrity of the presented work.
4 The paper discusses potential applications of the proposed method beyond trajectory generation, such as urban planning, transportation analysis, and location-based services. This discussion broadens the scope of the paper and highlights its practical relevance.


**Weaknesses:**

1 The paper could provide more in-depth discussions on the assumptions made by the diffusion probabilistic model and their potential impact on the generated trajectories. Understanding these assumptions is crucial for interpreting and contextualizing the results.
2 No clear instruction is provided on how the hyperparameters are selected for DiffTraj.


**Questions:**

1 How does the proposed diffusion probabilistic model capture both temporal and spatial dependencies in the trajectory generation process?
2 Can you elaborate on the generalizability of your approach to different geographic areas and datasets?
3 Can you explicitly state the limitations of your approach and discuss potential avenues for addressing them in future work? Acknowledging limitations and outlining future research directions would guide the community in further advancing trajectory generation methods.


**Limitations:**

The generalizability of the approach to diverse scenarios and datasets could be a limitation that warrants further exploration.

---

> ### Author Rebuttal · Authors · 2023-08-09
>
> We are delighted that the reviewer found our motivations and ideas interesting and original. Thank you for your positive opinions and insightful comments.
>
> > [W1]
>
> Thank you for highlighting the importance of discussing our diffusion probabilistic model's assumptions.
>
> 1. Noise assumption: we assume that the observed data is a noisy version of potentially clean data. This allows the model to reverse the noise addition process, aiming to recover the original noise-free trajectory. In this way, the quality and nature of the generated trajectories are influenced by the assumed noise distribution. If the real-world noise is significantly different from the assumed noise, this may result in discrepancies in the generated trajectories.
> 2. Non-equilibrium thermodynamic assumptions: In urban mobility, we assume that "particles" are individuals moving in an urban environment and that the "medium" symbolizes the urban space itself. Through a step-by-step denoising process, the disordered particles eventually follow specific paths in the medium, i.e., road trajectories in the city.
>
>
> > [W2]
>
> Thank you for pointing out the hyperparameter selection. We apologize for the oversight and appreciate the opportunity to clarify this aspect.
> The selection of hyperparameters for DiffTraj was guided by a combination of empirical experience and adherence to general settings commonly used in similar models and tasks. Here's how we approached it:
>
> 1. Empirical experience: We leveraged insights gained from previous work and experimentation to choose values (like input length) that align with the specific characteristics and requirements of our model.
> 2. General Settings: We also referred to general settings and best practices in the field, considering hyperparameters that have been shown to be effective in related models and tasks. This helped ensure that our choices were grounded in established knowledge and methodologies.
> 3. Iterative Refinement: While our initial selection was based on experience and general settings, we conducted iterative experimentation to fine-tune the hyperparameters. This process allowed us to identify the optimal combination that achieved the desired performance on our validation datasets.
>
> We recognize the initial manuscript's lack of detail in this area and will expand on our hyperparameter selection process in the revised version, detailing our choices, their rationale, and any guiding references.
>
> > [Q1]
>
> Thank you for your feedback. This is indeed a central aspect of our model, and I'm pleased to explain how it is achieved:
>
> 1. UNet structure: Individual Blocks in the UNet structure capture the relationships between consecutive points in the trajectory, essentially capturing spatial dependencies. Meanwhile, its multilevel structure and skip connections allow for the fusion of global and local contextual information to capture temporal dependencies.
> 2. External factor representation: The model represents external spatial-temporal dependencies through conditional embedding, where spatial (start/end area, travel distance, average distance) and temporal (departure time, travel time, average speed, etc.) features are encoded.
> 3. Diffusion process: By iterative denoising the data through a reverse denoising process, the model captures the underlying spatial structure of the trajectories. In addition, the training objective of the diffusion probabilistic model aims to minimize the differences between the generated trajectories and the real trajectories in both spatial and temporal dimensions. This ensures the model learns to capture and reproduce the spatial-temporal patterns inherent in the training data.
>
>
>
> > [Q2] & [Limitations]
>
> Thank you for highlighting the importance. We appreciate the opportunity to elaborate on this.
>
> DiffTraj is designed to capture patterns and behaviors inherent in trajectory data, which ensures the model can be adapted to different data sources and contexts. This means that the model can adapt to different geographic regions. Furthermore, Section 5.4 Transfer Learning in this paper validates this by showing that for a new city, only 5% of the data needs to be used for fine-tuning to enable the models to achieve significant performance in a new scenario. These results show that the DiffTraj model exhibits strong adaptability and generalization capabilities when applied to different urban scenarios.
> To further validate the generalizability of our approach, we conducted experiments on a different dataset **Porto** dataset.
>
> |Methods|Density|Trip|Length|Pattern|
>   |--|--|--|--|--|
>   |VAE       |0.0121|0.0224|0.0382|0.763|
>   |TrajGAN   |0.0101|0.0268|0.0332|0.765|
>   |Diffwave  |0.0106|0.0193|0.0266|0.799|
>   |Diff-LSTM |0.0092|0.0141|0.0255|0.828|
>   |DiffTraj  |0.0087|0.0132|0.0242|0.847|
>
> Through the new experiment, we can see that DiffTraj achieves the best performance on this dataset, demonstrating its ability to adapt to different geographic regions and capture different trajectory patterns. These results underscore the model's generalizability across different datasets.
>
> > [Q3]
>
> Thank you for emphasizing the need to address our approach's limitations and potential future directions. Here are the key points:
>
> 1. Our method relies on raw data for synthesis, ensuring that generated trajectories mirror real-world patterns. This dependency implies that the quality and characteristics of the raw data can influence the synthesized trajectories. While it also means that any biases or anomalies in the raw data might be reflected in the synthesized data.
> 2. While our trajectory generation is innovative, it's computationally demanding. Despite this, our approach still represents a more cost-effective solution than real-world data collection costs.
>
> While our work has limitations, it marks a notable step forward in trajectory generation. We are dedicated to refining our approach in future research, building on our established foundation.

---

> > ### Comment · Reviewer_gS1x · 2023-08-13
> > **Response to the Rebuttal**
> >
> > Thanks for the authors' detailed responses. The answers from the authors have fully addressed my questions about the paper. I have no more concerns with the technique. I am also content with the feedback to my question about "generalizability" that the authors supplement more datasets with good discussion. Taking all into consideration, I would raise my score. The information we discussed including the model clarification, generalizability, and potential future directions, are suggested to be put in a further version.

---

> > > ### Author Response · Authors · 2023-08-15
> > >
> > > We are truly grateful to the reviewer for taking the time to carefully assess our work and provide thoughtful feedback. Your suggestions, especially on model clarification and generalizability, have greatly improved our paper.

---

### Official Review · Reviewer_nB5g · 2023-07-09

**Soundness:** 3 good
**Presentation:** 3 good
**Contribution:** 2 fair
**Rating:** 4
**Confidence:** 4

**Summary:**

This paper adapts DDPM for trajectory generation within smart cities. The major contributions include the combination of different factors and the integration of some existing modules. The experiments over two real-world datasets can demonstrate the efficacy of the proposed model.

**Strengths:**

1. The paper is well-written and easy to follow.
2. The paper addresses an essential and important task in spatio-temporal data mining.
3. The experiments demonstrate the effectiveness of the proposed model over two real-world datasets.

**Weaknesses:**

1. Technical contribution is a bit weak. DDPM is popular in the era of AIGC. The paper adapts DDPM to trajectory generation but lacks novel or innovative designs. This drawback can be also seen in recent studies that borrow the idea of DDPMs to spatio-temporal data mining, e.g., DiffSTG [1]. By the way, this paper is very similar to DiffSTG, from model design to speed-up.
2. The paper suggests that incorporating the influence of external factors is challenging, but this claim may not be entirely supported. To the best of my knowledge, existing models can easily handle such influence, e.g., conditional GAN. As the proposed model simply integrates these factors as a condition in DDPM, it is unclear why capturing external factors would pose a significant challenge. Additionally, the paper lacks an ablation study that examines the impact of these external factors on the proposed model's performance.
3. To establish the generalizability of the proposed model, more real-world datasets from diverse applications and domains are needed. The exclusive use of datasets from DiDi in this paper limits the scope of the model's applicability and may not reflect its performance on other datasets.
4. The paper omits some important related work, such as [1, 2, 3]. Furthermore, [3] and [4, 5] (included in the paper) should be used as baselines for comparison to provide a comprehensive evaluation of the proposed model's performance.


Ref:

[1] Wen, Haomin, et al. "Diffstg: Probabilistic spatio-temporal graph forecasting with denoising diffusion models." arXiv preprint arXiv:2301.13629 (2023).

[2] Yuan, Yuan, et al. "Spatio-temporal Diffusion Point Processes." arXiv preprint arXiv:2305.12403 (2023).

[3] Feng, Jie, et al. "Learning to simulate human mobility." Proceedings of the 26th ACM SIGKDD international conference on knowledge discovery & data mining. 2020.

[4] Liu, Xi, Hanzhou Chen, and Clio Andris. "trajGANs: Using generative adversarial networks for geo-privacy protection of trajectory data (Vision paper)." Location privacy and security workshop. 2018.

[5] Zhang, Jing, et al. "Dp-trajgan: A privacy-aware trajectory generation model with differential privacy." Future Generation Computer Systems 142 (2023): 25-40.

**Questions:**

Please answer the questions in the weaknesses.

---

> ### Author Rebuttal · Authors · 2023-08-09
>
> We thank the reviewer for recognizing the importance of our work and for the well-written paper. We also appreciate the detailed comments posed by the reviewer. Please find below the point-to-point responses to the reviewer's comments.
>
> > [W1]
>
> Thank you for your feedback on our paper's technical contributions. While our work is indeed inspired by the success of DDPMs in generative models, adapting them to spatial-temporal trajectory data presented unique challenges due to differences like data dimensionality and temporal variations. Our Traj-UNet architecture is a testament to the innovations required to address these challenges.
> While both our work and DiffSTG  leverage DDPM-inspired models for the spatial-temporal domain, there are notable distinctions:
>
> 1. Purpose: DiffSTG focuses on traffic prediction, offering probabilistic traffic flow forecasts. In contrast, our work centers on trajectory generation, a generative task.
> 2. Performance: DiffSTG has limitations in prediction, whereas our model excels in trajectory generation, underscoring DDPMs' suitability for generative tasks.
> 3. Structural Designs: We explored various structures, including Wavenet and unconditional designs, eventually validating the efficacy of the Traj-UNet structure through rigorous experimentation.
>
> Our primary goal with DiffTraj was to address privacy concerns in GPS trajectory data. By melding DDPM strengths with trajectory-specific innovations, we've aimed to balance leveraging established methods and introducing novel techniques.
>
> > [W2]
>
> Thank you for your comment. While models like conditional GANs can incorporate external factors, the challenge we want to emphasize is the one we have in applying DDPM to trajectory generation. We'll clarify this distinction in our revised manuscript. Our model is designed not only to capture these interactions, but we can also use it as a guide for trajectory generation (We also present the corresponding experimental results in **Appendix C.2** and **Appendix C.3**).
>
> Regarding the ablation study, we've addressed this in the experimental section (**Table 1**). The performance metrics of **DiffTraj-wo/Con**, without external conditions, are notably inferior, underscoring the significance of these external factors. These factors not only enhance model performance but also guide trajectory generation. In Fig. 7 and Fig. 8 in the Appendix, we see that DiffTraj can specify the start and end areas of trajectories based on this. In addition, we can also customize the length, distance, travel time, etc., of the generated trajectories.
>
> >[W3]
>
> Thank you for pointing out the importance of the generalizability of our proposed model across diverse datasets. We wholeheartedly agree that relying solely on DiDi datasets could limit our model's perceived applicability. To address this concern, we have conducted additional experiments using the **Porto** dataset, allowing us to test our model's robustness and adaptability further. The main experimental results and metrics on this dataset perform as follows:
>
> |Methods|Density|Trip|Length|Pattern|
>   |--|--|--|--|--|
>   |VAE       |0.0121|0.0224|0.0382|0.763|
>   |TrajGAN   |0.0101|0.0268|0.0332|0.765|
>   |Diffwave  |0.0106|0.0193|0.0266|0.799|
>   |Diff-LSTM |0.0092|0.0141|0.0255|0.828|
>   |DiffTraj  |0.0087|0.0132|0.0242|0.847|
>
> Our model still achieves superior performance on the new dataset, which is comparable to its performance on the DiDi datasets. These results reinforce the generalizability of our model across different datasets and domains. We believe that including the **Porto** dataset and the DiDi datasets provides a more comprehensive evaluation of our model's performance and applicability.
>
> > [W4]
>
> Thank you for your feedback. We will revisit the works you mentioned ([1, 2, 3]) and acknowledge their relevance to our study. We will incorporate a discussion of these works, highlighting their contributions and differentiating our approach from theirs. This will provide readers with a more complete understanding of the state of the art and the novelty of our proposed model.
>
> We appreciate your suggestion to use [3] and [4, 5] as baselines for comparison. Including these works in our evaluation will indeed provide a more comprehensive assessment of our model's performance.
> Among them, the TrajGAN in our work is exactly the model proposed in the literature [4]. The main difference between the literature [4] and the literature [5] is that the former uses CNNs, and the latter uses LSTMs. In this regard, we have reproduced both of them, and the results are as follows:
>
> ||**Chengdu**||||**Xi'an**||||
> |--|--|--|--|--|--|--|--|--|
> ||**Density**|**Trip**|**Length**|**Pattern**|**Density**|**Trip**|**Length**|**Pattern**|
> | TrajGAN [4]    |0.0125|0.0497|0.0388|0.502 |0.0220|0.0512|0.0386 | 0.565|
> | DP-TrajGAN [5] |0.0117|0.0443|0.0221|0.706 |0.0207|0.0498|0.0436 | 0.664|
>
> However, we want to clarify the core differences between our approach and the methodology in [3]. In [3], the trajectory data is converted into a grid-based representation, and they focus on simulating human behavior by predicting these grids. This grid-based approach inherently changes the nature of the data and the problem, making it a distinct task from the continuous trajectory generation in our work. Our model is designed to handle and generate exact trajectory data without converting it into a discrete grid format. Given these foundational disparities, a direct comparison might not be meaningful.
>
> Nevertheless, we value the contribution of [3] to human mobility simulation and will highlight its significance while distinguishing our methodology in the revised manuscript. We trust this clarifies our stance and appreciate your feedback.

---

### Author Rebuttal · Authors · 2023-08-09

We appreciate the insightful comments and perspectives of the reviewers, and the attached figures and tables of results are included as a supplement **PDF** to the rebuttal.

---

### Decision · Program_Chairs · 2023-09-21

**Decision:**

Accept (poster)

**Comment:**

Initially, the paper raised several concerns:

Privacy Claims and Lack of Evidence: Although the paper prominently claims to prioritize privacy, there was a distinct lack of empirical support to validate these claims. While the authors demonstrated that training on their generated data produced results similar to the original data, there was an underlying concern about potential data copying by the generative model.

Dependence on Original Data: The necessity to rely on the original data to train the generative model posed questions about the genuine advantages and novelty of using the generated data.

Weak Comparisons: The benchmarking seemed inadequate. Relying mostly on self-comparisons or using outdated models for evaluation diminishes the paper's value as it doesn't offer a comprehensive view of its standing within current advancements in the field.

However, post the initial review:

Rebuttal Improvements: The authors have made commendable efforts to address the concerns raised during the initial review. They conducted additional experiments which, evidently, clarified and resolved some of the initial apprehensions. This proactive approach has led to a noticeable shift in the perception of the paper, with many reviewers now providing more favorable ratings.

The increased ratings from the reviewers post-rebuttal indicate a positive shift, but it's essential to ensure that all significant concerns will be thoroughly addressed in the final version.